# Estimating decision tree learnability
# with polylogarithmic sample complexity

**Guy Blanc**
Stanford University
gblanc@cs.stanford.edu

**Neha Gupta**
Stanford University
nehagupta@cs.stanford.edu

**Jane Lange**
Massachusetts Institute of Technology
jlange@mit.edu

**Li-Yang Tan**
Stanford University
liyang@cs.stanford.edu

## Abstract

We show that top-down decision tree learning heuristics (such as ID3, C4.5, and CART) are amenable to highly efficient *learnability estimation*: for monotone target functions, the error of the decision tree hypothesis constructed by these heuristics can be estimated with *polylogarithmically* many labeled examples, exponentially smaller than the number necessary to run these heuristics, and indeed, exponentially smaller than information-theoretic minimum required to learn a good decision tree. This adds to a small but growing list of fundamental learning algorithms that have been shown to be amenable to learnability estimation.

En route to this result, we design and analyze sample-efficient *minibatch* versions of top-down decision tree learning heuristics and show that they achieve the same provable guarantees as the full-batch versions. We further give "active local" versions of these heuristics: given a test point $x^\star$, we show how the label $T(x^\star)$ of the decision tree hypothesis $T$ can be computed with polylogarithmically many labeled examples, exponentially smaller than the number necessary to learn $T$.

## 1 Introduction

We study the problem of *estimating learnability*, recently introduced by Kong and Valiant [KV18] and Blum and Hu [BH18]. Consider a learning algorithm $\mathcal{A}$ and a dataset $S$ of *unlabeled* examples. Can we estimate the performance of $\mathcal{A}$ on $S$—that is, the error of the hypothesis that $\mathcal{A}$ would return if we were to label the entire dataset $S$ and train $\mathcal{A}$ on it—by labeling only very few of the examples in $S$? Are there learning tasks and algorithms for which an accurate estimate of learnability can be obtained with far fewer labeled examples than the information-theoretic minimum required to learn a good hypothesis?

**Motivating applications.** Across domains and applications, the labeling of datasets is often an expensive process, requiring either significant computational resources or a large number of person-hours. There are therefore numerous natural scenarios in which an efficient learnability estimation procedure could serve as a useful exploratory precursor to learning. For example, suppose the error estimate returned by this procedure is large. This tells us that if we were to label the entire dataset $S$ and run $\mathcal{A}$ on it, the error of the hypothesis $h$ that $\mathcal{A}$ would return is large. With this information, we may decide that $h$ would not have been of much utility anyway, thereby saving ourselves the resources and effort to label the entire dataset $S$ (and to run $\mathcal{A}$). Alternatively, we may decide to collect more data or to enlarge the feature space of $S$, in hopes of improving the performance of $\mathcal{A}$. The learnability estimation procedure could again serve as a guide in this process, telling us *how*

*much* the performance of $\mathcal{A}$ would improve with these decisions. Relatedly, such a procedure could be useful for hyperparameter tuning, where the learning algorithm $\mathcal{A}$ takes as input a parameter $\rho$, and its performance improves with $\rho$, but its time and sample complexity also increases with $\rho$. The learnability estimation procedure enables us to efficiently determine the best choice of $\rho$ for our application at hand, and run $\mathcal{A}$ just a single time with this value of $\rho$. As a final example, such a procedure could also be useful for dataset selection: given unlabeled training sets $S_1, \ldots, S_m$, and access to labeled examples from a test distribution $\mathcal{D}$, we can efficiently determine the $S_i$ for which $\mathcal{A}$ would produce a hypothesis that achieves the smallest error with respect to $\mathcal{D}$.

**Prior works on estimating learnability.** While this notion is still relatively new, there are by now a number of works studying it in a variety of settings, including robust linear regression [KV18], learning unions of intervals and $k$-Nearest-Neighbor algorithms [BH18], contextual bandits [KVB20], learning Lipschitz functions, and the Nadaraya–Watson estimator in kernel regression [BBG20]. A striking conceptual message has emerged from this line of work: it is often possible to estimate learnability with far fewer labeled examples than the number required to run the corresponding algorithm, and indeed, far fewer than the information-theoretic minimum required to learn a good hypothesis.

## 1.1 Top-down decision tree learning

We study the problem of estimating learnability in the context of *decision tree learning*. Specifically, we focus on *top-down* decision tree learning heuristics such as ID3 [Qui86], C4.5 [Qui93], and CART [Bre17]. These classic and simple heuristics continue to be widely employed in everyday machine learning applications and enjoy significant empirical success. They are also the core subroutine in modern, state-of-the-art ensemble methods such as random forests [Bre01] and gradient boosted trees [CG16].

We briefly describe how these top-down heuristics work, deferring the formal description to the main body of this paper. Each such heuristic TOPDOWN$_\mathscr{G}$ is defined by *impurity function* $\mathscr{G}$ : $[0, 1] \to [0, 1]$ which determines its splitting criterion.[1] TOPDOWN$_\mathscr{G}$ takes as input a labeled dataset $S \subseteq \mathcal{X} \times \{0, 1\}$ and a size parameter $t \in \mathbb{N}$, and constructs a size-$t$ decision tree for $S$ in a *greedy, top-down* fashion. It begins by querying $\mathbb{1}[x_i \geq \theta]$ at the root of the tree, where $x_i$ and $\theta$ are chosen to maximize the *purity gain with respect to* $\mathscr{G}$:

$$\mathscr{G}(\mathbb{E}[\boldsymbol{y}]) - \big( \Pr[\boldsymbol{x}_i \geq \theta] \cdot \mathscr{G}(\mathbb{E}[\,\boldsymbol{y} \mid \boldsymbol{x}_i \geq \theta\,]) + \Pr[\boldsymbol{x}_i < \theta] \cdot \mathscr{G}(\mathbb{E}[\,\boldsymbol{y} \mid \boldsymbol{x}_i < \theta\,])\big),$$

where the expectations and probabilities are with respect to $(\boldsymbol{x}, \boldsymbol{y}) \sim S$. More generally, TOPDOWN$_\mathscr{G}$ grows its current tree $T$ by splitting a leaf $\ell \in T^\circ$ with a query to $\mathbb{1}[x_i \geq \theta]$, where $\ell$, $x_i$, and $\theta$ are chosen to maximize:

$$\mathrm{PurityGain}_{\mathscr{G}, S}(\ell, i, \theta) \coloneqq \Pr[\boldsymbol{x} \text{ reaches } \ell] \cdot \mathrm{LocalGain}_{\mathscr{G}, S}(\ell, i, \theta),$$

where

$$\begin{aligned}
\mathrm{LocalGain}_{\mathscr{G}, S}(\ell, i, \theta) \coloneqq {} & \mathscr{G}(\mathbb{E}[\,\boldsymbol{y} \mid \boldsymbol{x} \text{ reaches } \ell\,]) \\
& - \big( \Pr[\boldsymbol{x}_i \geq \theta] \cdot \mathscr{G}(\mathbb{E}[\,\boldsymbol{y} \mid \boldsymbol{x} \text{ reaches } \ell, \boldsymbol{x}_i \geq \theta\,]) \\
& + \Pr[\boldsymbol{x}_i < \theta] \cdot \mathscr{G}(\mathbb{E}[\,\boldsymbol{y} \mid \boldsymbol{x} \text{ reaches } \ell, \boldsymbol{x}_i < \theta\,])\big).
\end{aligned}$$

**Provable guarantees for monotone target functions [BLT20a].** Motivated by the tremendous popularity and empirical successes of these top-down heuristics, there has been significant interest and efforts in establishing provable guarantees on their performance [Kea96, DKM96, KM99, FP04, Lee09, BDM19b, BDM19a, BLT20b, BLT20a]. The starting point of our work is a recent result of Blanc et al. [BLT20a], which provides a guarantee on their performance when run on *monotone* target functions, with respect to the uniform distribution:

**Theorem** (Theorem 2 of [BLT20a]). *Let $f : \{\pm 1\}^d \to \{0,1\}$ be a monotone target function and $\mathscr{G}$ be any impurity function. For $s \in \mathbb{N}$ and $\varepsilon, \delta \in (0, \frac{1}{2})$, let $t = s^{\Theta(\log s)/\varepsilon^2}$ and $\boldsymbol{S}$ be a set of $n$ labeled training examples $(\boldsymbol{x}, f(\boldsymbol{x}))$ where $\boldsymbol{x} \sim \{\pm 1\}^d$ is uniform random, and $n = \tilde{O}(t) \cdot \mathrm{poly}(\log d, \log(1/\delta))$.*

*With probability at least $1 - \delta$ over the randomness of $\boldsymbol{S}$, the size-$t$ decision tree hypothesis constructed by $\mathrm{TOPDOWN}_{\mathscr{G}}(t, \boldsymbol{S})$ satisfies $\mathrm{error}_f(T) := \mathrm{Pr}_{\boldsymbol{x} \sim \{\pm 1\}^d}[T(\boldsymbol{x}) \neq f(\boldsymbol{x})] \leq \mathsf{opt}_s + \varepsilon$, where $\mathsf{opt}_s$ denotes the error of the best size-$s$ decision tree for $f$.*

We refer the reader to the introduction of [BLT20a] for a discussion of why assumptions on the target function are necessary in order to establish provable guarantees. Briefly, as had been noted by Kearns [Kea96], there are examples of simple non-monotone target functions $f : \{\pm 1\}^d \to \{0, 1\}$, computable by decision trees of constant size, for which any impurity-based heuristic may build a complete tree of size $\Omega(2^d)$ before achieving any non-trivial accuracy. Monotonicity is a natural way of excluding these adversarial functions, and for this reason it is one of the most common assumptions in learning theory. Results for monotone functions tend to be good proxies for the performance of learning algorithms on real-world datasets, which also do not exhibit these adversarial structures.

**Our contributions.** We give strengthened provable guarantees on the performance of top-down decision tree learning heuristics, focusing on sample complexity. Our three main contributions are as follows:

1. *Minibatch top-down decision tree learning.* We introduce and analyze $\mathrm{MINIBATCHTOPDOWN}_{\mathscr{G}}$, a *minibatch* version of $\mathrm{TOPDOWN}_{\mathscr{G}}$ where the purity gain associated with each split is estimated with only polylogarithmically many samples within the dataset $S$ rather than all of $S$. For all impurity functions $\mathscr{G}$, we show that $\mathrm{MINIBATCHTOPDOWN}_{\mathscr{G}}$ achieves the same provable guarantees that those that [BLT20a] had established for the full-batch version $\mathrm{TOPDOWN}_{\mathscr{G}}$.

2. *Active local learning.* We then study $\mathrm{MINIBATCHTOPDOWN}_{\mathscr{G}}$ within the recently-introduced *active local learning* framework of Backurs, Blum, and Gupta [BBG20], and show that it admits an efficient active local learner. Given active access to an *unlabeled* dataset $S$ and a test point $x^\star$, we show how $T(x^\star)$ can be computed by labeling only polylogarithmically many of the examples in $S$, where $T$ is the decision tree hypothesis that $\mathrm{MINIBATCHTOPDOWN}_{\mathscr{G}}$ would construct if we were to label *all* of $S$ and train $\mathrm{MINIBATCHTOPDOWN}_{\mathscr{G}}$ on it.

3. *Estimating learnability.* Building on both our results above, we show that $\mathrm{MINIBATCHTOPDOWN}_{\mathscr{G}}$ is amendable to highly-efficient learnability estimation. Given active access to an unlabeled dataset $S$, we show that the error of $T$ with respect to any test distribution can be approximated by labeling only polylogarithmically many of the examples in $S$, where $T$ is the decision tree hypothesis that $\mathrm{MINIBATCHTOPDOWN}_{\mathscr{G}}$ would construct if we were to label all of $S$ and train $\mathrm{MINIBATCHTOPDOWN}_{\mathscr{G}}$ on it.

### 1.2 Formal statements of our results

**Feature space and distributional assumptions.** We work in the setting of binary attributes and binary classification, i.e. we focus on the task of learning a target function $f : \{\pm 1\}^d \to \{0, 1\}$. We will assume the learning algorithm receives uniform random examples $\boldsymbol{x} \sim \{\pm 1\}^d$, either labeled or unlabeled. The error of a decision tree hypothesis $T : \{\pm 1\}^d \to \{0, 1\}$ with respect to $f$ is defined to be $\mathrm{error}_f(T) := \mathrm{Pr}[f(\boldsymbol{x}) \neq T(\boldsymbol{x})]$ where $\boldsymbol{x} \sim \{\pm 1\}^d$ is uniform random. We write $\mathsf{opt}_s(f)$ to denote $\min\{\mathrm{error}_f(T) : T \text{ is a size-}s \text{ decision tree}\}$; when $f$ is clear from context we simply write $\mathsf{opt}_s$. We will also be interested in the error of $T$ with respect to general test sets $(\mathrm{Pr}_{(\boldsymbol{x}, \boldsymbol{y}) \sim S_{\text{test}}}[T(\boldsymbol{x}) \neq \boldsymbol{y}])$ and general test distributions $(\mathrm{Pr}_{(\boldsymbol{x}, \boldsymbol{y}) \sim \mathcal{D}_{\text{test}}}[T(\boldsymbol{x}) \neq \boldsymbol{y}])$.

**Notation and terminology.** For any decision tree $T$, we say the size of $T$ is the number of leaves in $T$. We refer to a decision tree with unlabeled leaves as a *partial tree*, and write $T^\circ$ to denote such trees. For a leaf $\ell$ of a partial tree $T^\circ$, we write $|\ell|$ to denote its depth within $T^\circ$, the number of attributes queried along the path that leads to $\ell$. We say that an input $x \in \{\pm 1\}^d$ is *consistent with a leaf $\ell$* if $x$ reaches $\ell$ within $T^\circ$, and we write $\ell_{T^\circ}(x)$ to denote the (unique) leaf $\ell$ of $T^\circ$ that $x$ is consistent with. A function $f : \{\pm 1\}^d \to \{0, 1\}$ is said to be *monotone* if for every coordinate $i \in [d]$, it is either non-decreasing with respect to $i$ (i.e. $f(x) \leq f(y)$ for all $x, y \in \{\pm 1\}^d$ such

that $x_i \leq y_i$) or non-increasing with respect to $i$ (i.e. $f(x) \geq f(y)$ for all $x, y \in \{\pm 1\}^d$ such that $x_i \leq y_i$).

We use **boldface** to denote random variables (e.g. $\boldsymbol{x} \sim \{\pm 1\}^d$), and unless otherwise stated, all probabilities and expectations are with respect to the uniform distribution. For $p \in [0, 1]$, we write $\mathrm{round}(p)$ to denote $\mathbb{1}[p \geq \frac{1}{2}]$. We reserve $S$ to denote a labeled dataset and $S^\circ$ to denote an unlabeled dataset.

We are now ready to describe new algorithms and state our main results.

**Definition 1** (Minibatch). *Let $S$ be a labeled dataset. A* minibatch *from $S$, denoted $\boldsymbol{B} \sim \mathrm{Batch}_b(S)$, is a set of $b$ uniform random points $(x, y)$ chosen without replacement from $S$. More generally, for a leaf $\ell$, a* minibatch consistent with $\ell$ *from $S$, denoted $\boldsymbol{B} \sim \mathrm{Batch}_b(S, \ell)$, is a set of $b$ uniformly random pairs chosen without replacement from among $(x, y) \in S$ such that $x$ is consistent with $\ell$. (In both cases, if there are fewer than $b$ such points, we return all of them.) Minibatches from unlabeled datasets $S^\circ$ are defined analogously.*

**Definition 2** (Minibatch completion of partial trees). *Given a partial tree $T^\circ$, we write $T^\circ_{\mathrm{Batch}_b(S)}$ to denote the tree obtained by labeling each leaf $\ell \in T^\circ$ with $\mathrm{round}(\mathbb{E}_{(\boldsymbol{x}, f(\boldsymbol{x})) \sim \boldsymbol{B}}[f(\boldsymbol{x})])$ where $\boldsymbol{B} \sim \mathrm{Batch}_b(S, \ell)$.*

---

$\mathrm{MINIBATCHTOPDOWN}_{\mathscr{G}}(t, b, S)$:

    Initialize $T^\circ$ to be the empty tree.

    Define $D := \log t + \log \log t$.

    while ($\mathrm{size}(T^\circ) < t$) {

       1. *Score:* For each leaf $\ell \in T^\circ$ of depth at most $D$, draw $\boldsymbol{B} \sim \mathrm{Batch}_b(S, \ell)$. For each coordinate $i \in [d]$, compute:

$$\mathrm{PurityGain}_{\mathscr{G}, \boldsymbol{B}}(\ell, i) := 2^{-|\ell|} \cdot \mathrm{LocalGain}_{\mathscr{G}, \boldsymbol{B}}(\ell, i), \text{ where}$$
$$\mathrm{LocalGain}_{\mathscr{G}, \boldsymbol{B}}(\ell, i) := \mathscr{G}(\mathbb{E}[f(\boldsymbol{x})])$$
$$- \left( \tfrac{1}{2} \mathscr{G}(\mathbb{E}[\, f(\boldsymbol{x}) \mid \boldsymbol{x}_i = -1\,]) + \tfrac{1}{2} \mathscr{G}(\mathbb{E}[\, f(\boldsymbol{x}) \mid \boldsymbol{x}_i = 1\,]) \right),$$

         where the expectations are with respect to $(\boldsymbol{x}, f(\boldsymbol{x})) \sim \boldsymbol{B}$.

       2. *Split:* Let $(\ell^\star, i^\star)$ be the tuple that maximizes $\mathrm{PurityGain}_{\mathscr{G}, \boldsymbol{B}}(\ell, i)$. Grow $T^\circ$ by splitting $\ell^\star$ with a query to $x_{i^\star}$.

    }

    Output $T^\circ_{\mathrm{Batch}_b(S)}$.

---

Figure 1: $\mathrm{MINIBATCHTOPDOWN}_{\mathscr{G}}$ takes as input a size parameter $t$, a minibatch size $b$, and a labeled dataset $S$. It outputs a size-$t$ decision tree hypothesis for $f$.

$\mathrm{MINIBATCHTOPDOWN}_{\mathscr{G}}$ is a minibatch version of $\mathrm{TOPDOWN}_{\mathscr{G}}$, which we described informally in Section 1.1 and include its full pseudocode in Section 5. $\mathrm{MINIBATCHTOPDOWN}_{\mathscr{G}}$ is more efficient than $\mathrm{TOPDOWN}_{\mathscr{G}}$ in two respects: first, purity gains and completions are computed with respect to a minibatch $\boldsymbol{B}$ of size $b$ instead of all the entire dataset $S$; second, $\mathrm{MINIBATCHTOPDOWN}_{\mathscr{G}}$ never splits a leaf of depth greater than $D$, and hence constructs a decision tree of small size *and* small depth, rather than just small size. (Looking ahead, both optimizations will be crucial for the design of our sample-efficient active local learning and learnability estimation procedures.)

Our first result shows that $\mathrm{MINIBATCHTOPDOWN}_{\mathscr{G}}$ achieves the same performance guarantees as those that [BLT20a] had established for the full-batch version $\mathrm{TOPDOWN}_{\mathscr{G}}$:

**Theorem 1** (Provable guarantees for $\mathrm{MINIBATCHTOPDOWN}$; informal version). *Let $f : \{\pm 1\}^d \to \{0, 1\}$ be a monotone target function and fix an impurity function $\mathscr{G}$. For any $s \in \mathbb{N}$, $\varepsilon, \delta \in (0, \frac{1}{2})$, let $t = s^{\Theta(\log s)/\varepsilon^2}$, and $\boldsymbol{S}$ be a set of $n$ labeled training examples $(\boldsymbol{x}, f(\boldsymbol{x}))$ where $\boldsymbol{x} \sim \{\pm 1\}^d$ is uniform random, and*

$$n = \tilde{O}(t) \cdot \mathrm{poly}(\log d, \log(1/\delta)).$$

*If the minibatch size is at least*

$$b = \mathrm{polylog}(t) \cdot \mathrm{poly}(\log d, \log(1/\delta)),$$

*then with probability at least $1 - \delta$ over the randomness of $\boldsymbol{S}$ and the draws of minibatches from within $\boldsymbol{S}$, the size-$t$ decision tree hypothesis constructed by* $\mathrm{MiniBatchTopDown}_\mathscr{G}(t, b, \boldsymbol{S})$ *satisfies* $\mathrm{error}_f(T) \leq \mathsf{opt}_s + \varepsilon$.

Theorem 1 shows that it suffices for the minibatch size $b$ of $\mathrm{MiniBatchTopDown}_\mathscr{G}$ to depend polylogarithmically on $t$; in contrast, the full-batch version $\mathrm{TopDown}_\mathscr{G}$ uses the entire set $S$ to compute purity gains and determine its splits, and $|S| = n$ has a superlinear dependence on $t$.

Our next algorithm is an implementation of $\mathrm{MiniBatchTopDown}_\mathscr{G}$ within the *active local learning* framework of Backurs, Blum, and Gupta [BBG20]:

---

$\mathrm{LocalLearner}_\mathscr{G}(t, b, S^\circ, x^\star)$:

Initialize $T^\circ$ to be the empty tree.

Define $D := \log t + \log \log t$.

Initialize $e := 1$ and let $\boldsymbol{B}^\circ_{\mathrm{strands}}$ be $b$ uniform random points from $\{\pm 1\}^d$.

while ($e < t$) {

1. *Score:* For each leaf $\ell \in \{\ell_{T^\circ}(x) : x \in \boldsymbol{B}^\circ_{\mathrm{strands}} \cup \{x^\star\}\}$ of depth at most $D$, draw $\boldsymbol{B}^\circ \sim \mathrm{Batch}_b(S^\circ, \ell)$, query $f$'s values on these points. For each coordinate $i \in [d]$, compute:

$$\mathrm{PurityGain}_{\mathscr{G}, \boldsymbol{B}^\circ}(\ell, i) := 2^{-|\ell|} \cdot \mathrm{LocalGain}_{\mathscr{G}, \boldsymbol{B}^\circ}(\ell, i), \text{ where}$$

$$\mathrm{LocalGain}_{\mathscr{G}, \boldsymbol{B}^\circ}(\ell, i) := \mathscr{G}(\mathbb{E}[f(\boldsymbol{x})])$$
$$- \left( \tfrac{1}{2} \mathscr{G}(\mathbb{E}[\, f(\boldsymbol{x}) \mid \boldsymbol{x}_i = -1\,]) + \tfrac{1}{2} \mathscr{G}(\mathbb{E}[\, f(\boldsymbol{x}) \mid \boldsymbol{x}_i = 1\,]) \right),$$

where the expectations are with respect to $\boldsymbol{x} \sim \boldsymbol{B}^\circ$.

2. *Split:* Let $(\ell^\star, i^\star)$ be the tuple that maximizes $\mathrm{PurityGain}_{\mathscr{G}, \boldsymbol{B}^\circ}(\ell, i)$. Grow $T^\circ$ by splitting $\ell^\star$ with a query to $x_{i^\star}$.

3. *Estimate size:* Update our size estimate to

$$\boldsymbol{e} = \mathop{\mathbb{E}}_{\boldsymbol{x} \sim \boldsymbol{B}^\circ_{\mathrm{strands}}} \left[ 2^{|\ell_{T^\circ}(\boldsymbol{x})|} \right].$$

}

Draw $\boldsymbol{B}^\circ \sim \mathrm{Batch}_b(S^\circ, \ell_{T^\circ}(x^\star))$ and query $f$'s values on these points.

Output $\mathrm{round}(\mathbb{E}_{\boldsymbol{x} \sim \boldsymbol{B}^\circ}[f(\boldsymbol{x})])$.

---

Figure 2: $\mathrm{LocalLearner}_\mathscr{G}$ takes as input a size parameter $t$, a minibatch size $b$, an *unlabeled* dataset $S^\circ$, and an input $x^\star$. It selectively queries $f$'s values on a few points within $S^\circ$ and outputs $T(x^\star)$, where $T$ is a tree of size approximately $t$ that $\mathrm{MiniBatchTopDown}_\mathscr{G}$ would return if we were to label all of $S^\circ$ and train $\mathrm{MiniBatchTopDown}_\mathscr{G}$ on it.

**Theorem 2** (Active local version of $\mathrm{MiniBatchTopDown}$; informal version)**.** *Let $f : \{\pm 1\}^d \to \{0, 1\}$ be a target function, $\mathscr{G}$ be an impurity function, and $S^\circ$ be an unlabeled training set.*

*For all $t \in \mathbb{N}$, $\eta, \delta \in (0, \frac{1}{2})$, if the minibatch size is at least $b = \mathrm{poly}(\log t, \log d, 1/\eta, \log(1/\delta))$, then with probability at least $1 - \delta$ over the randomness of $\boldsymbol{B}^\circ_{\mathrm{strands}}$, we have that for all $x^\star \in \{\pm 1\}^d$, $\mathrm{LocalLearner}_\mathscr{G}(t, b, S^\circ, x^\star)$ labels*

$$q = O(b^2 \log t) = \mathrm{polylog}(t) \cdot \mathrm{poly}(\log d, 1/\eta, \log(1/\delta))$$

*points within $S^\circ$ and returns $T(x^\star)$, where $T$ is the size-$t'$ decision tree hypothesis that $\mathrm{MiniBatchTopDown}_\mathscr{G}(t', b, S)$ would construct, $t' \in t(1 \pm \eta)$, and $S$ is the labeled dataset obtained by labeling all of $S^\circ$ with $f$'s values.[2]*

Theorem 2 yields, as a fairly straightforward consequence, our learnability estimation procedure $\text{Est}_{\mathscr{G}}$ that estimates the performance of MINIBATCHTOPDOWN$_{\mathscr{G}}$ with respect to any test set $S_{\text{test}}$:

**Theorem 3** (Estimating learnability of MINIBATCHTOPDOWN; informal version)**.** *Let $f : \{\pm 1\}^d \to \{0, 1\}$ be a target function, $\mathscr{G}$ be an impurity function, $S^\circ$ be an unlabeled training set, and $S_{\text{test}}$ be a labeled test set.*

*For all $t \in \mathbb{N}$ and $\eta, \delta \in (0, \frac{1}{2})$, if the minibatch size $b$ is as in Theorem 2, then with probability at least $1 - \delta$ over the randomness of the draws of minibatches from within $S^\circ$, $\text{EST}_{\mathscr{G}}(t, b, S^\circ, S_{\text{test}})$ labels*

$$q = O(|S_{\text{test}}| \cdot b \log t + b^2 \log t) = |S_{\text{test}}| \cdot \text{polylog}(t) \cdot \text{poly}(\log d, 1/\eta, \log(1/\delta))$$

*points within $S^\circ$ and returns the error of $T$ with respect to $S_{\text{test}}$,*

$$\text{error}_{S_{\text{test}}}(T) \coloneqq \Pr_{(\boldsymbol{x}, \boldsymbol{y}) \sim S_{\text{test}}} [T(\boldsymbol{x}) \neq \boldsymbol{y}],$$

*where $T$ is as in Theorem 2.*

We remark that Theorem 1 requires the training set be composed of independent draws of $(\boldsymbol{x}, f(\boldsymbol{x}))$ where $\boldsymbol{x}$ is drawn uniformly from $\{\pm 1\}^d$. On the other hand, in Theorems 2 and 3, the high probability guarantees hold for any fixed choice of training set $S^\circ$. Similarly, in Theorem 3, $S_{\text{test}}$ can be arbitrarily chosen. Indeed, as an example application of Theorem 3, we can let $\boldsymbol{S}_{\text{test}}$ be $\Theta(\log(1/\delta)/\varepsilon^2)$ many labeled examples $(\boldsymbol{x}, \boldsymbol{y})$ drawn from an arbitrary test distribution $\mathcal{D}_{\text{test}}$ over $\{\pm 1\}^d \times \{0, 1\}$, where the marginal over $\{\pm 1\}^d$ need not be uniform and the the labels need not be consistent with $f$. With probability at least $1 - \delta$, the output of $\text{Est}_{\mathscr{G}}$ will be within $\pm \varepsilon$ of $\Pr_{(\boldsymbol{x}, \boldsymbol{y}) \sim \mathcal{D}_{\text{test}}}[T(\boldsymbol{x}) \neq \boldsymbol{y}]$.

## 2 Proof overview for Theorem 1

Our proof of Theorem 1 builds upon and extends the analysis in [BLT20a]. (Recall that [BLT20a] analyzed the full-batch version TOPDOWN$_{\mathscr{G}}$, which we have included in Section 5 of this paper, and their guarantee concerning its performance is their Theorem 2, which we have stated in Section 1.1 of this paper). In this section we give a high-level overview of both [BLT20a]'s and our proof strategy, in tandem with a description of the technical challenges that arise as we try to strengthen [BLT20a]'s Theorem 2 to our Theorem 1.

Let $f : \{\pm 1\}^d \to \{0, 1\}$ be a monotone function and fix an impurity function $\mathscr{G}$. Let $T^\circ$ be a partial tree that is being built by either TOPDOWN$_{\mathscr{G}}$ or MINIBATCHTOPDOWN$_{\mathscr{G}}$. Recall that TOPDOWN$_{\mathscr{G}}$ and MINIBATCHTOPDOWN$_{\mathscr{G}}$ compute, for each leaf $\ell \in T^\circ$ and coordinate $i \in [d]$, $\text{PurityGain}_{\mathscr{G}, S}(\ell, i)$ and $\text{PurityGain}_{\mathscr{G}, \boldsymbol{B}}(\ell, i)$ respectively. Both these quantities can be thought of as estimates of the *true* purity gain:

$$\text{PurityGain}_{\mathscr{G}, f}(\ell, i) \coloneqq 2^{-|\ell|} \cdot \text{LocalGain}_{\mathscr{G}, f}(\ell, i) \text{ where}$$
$$\text{LocalGain}_{\mathscr{G}, f}(\ell, i) \coloneqq \mathscr{G}(\mathbb{E}[\, f(\boldsymbol{x}) \mid \boldsymbol{x} \text{ reaches } \ell\,])$$
$$- \left( \tfrac{1}{2} \mathscr{G}(\mathbb{E}[\, f(\boldsymbol{x}) \mid \boldsymbol{x} \text{ reaches } \ell, \boldsymbol{x}_i = -1\,]) \right.$$
$$+ \tfrac{1}{2} \mathscr{G}(\mathbb{E}[\, f(\boldsymbol{x}) \mid \boldsymbol{x} \text{ reaches } \ell, \boldsymbol{x}_i = 1\,])),$$

where here and throughout this section, all expectations are with respect to a uniform random $\boldsymbol{x} \sim \{\pm 1\}^d$. The fact that MINIBATCHTOPDOWN$_{\mathscr{G}}$'s estimates of this true purity gain are based on minibatches $\boldsymbol{B}$ of size exponentially smaller than that of the full sample set $S$—and hence could be exponentially less accurate—is a major source of technical challenges that arise in extending [BLT20a]'s guarantees for TOPDOWN$_{\mathscr{G}}$ to MINIBATCHTOPDOWN$_{\mathscr{G}}$.

[BLT20a] considers the potential function:

$$\mathscr{G}\text{-impurity}_f(T^\circ) \coloneqq \sum_{\text{leaves } \ell \,\in\, T^\circ} 2^{-|\ell|} \cdot \mathscr{G}(\mathbb{E}[f_\ell]).$$

The following fact about this potential function $\mathscr{G}\text{-impurity}_f$ is straightforward to verify (and is proved in [BLT20a]):

---

Similarly, if one then wished to actually construct this tree $T$, they would run MINIBATCHTOPDOWN$_{\mathscr{G}}$ with these same outcomes of randomness.

**Fact 2.1.** *For any partial tree $T^\circ$, leaf $\ell \in T^\circ$, and coordinate $i \in [d]$, let $\tilde{T}^\circ$ be the tree obtained from $T^\circ$ by splitting $\ell$ with a query to $x_i$. Then,*

$$\mathscr{G}\text{-impurity}_f(\tilde{T}^\circ) = \mathscr{G}\text{-impurity}_f(T^\circ) - \text{PurityGain}_{\mathscr{G},f}(\ell, i).$$

A key ingredient in [BLT20a]'s analysis is a proof that as long as $\text{error}_f(T_S^\circ) > \text{opt}_s + \varepsilon$ (where $T_S^\circ$ denotes the completion of $T^\circ$ with respect to the full batch $S$; see Section 5), there must be a leaf $\ell \in T^\circ$ and coordinate $i$ with high true purity gain, $\text{PurityGain}_{\mathscr{G},f}(\ell, i) \geq \text{poly}(\varepsilon/t)$. Since $\text{TOPDOWN}_{\mathscr{G}}$'s estimates $\text{PurityGain}_{\mathscr{G},S}$ of $\text{PurityGain}_{\mathscr{G},f}$ are with respect to a sample of size $|S| \geq \text{poly}(t/\varepsilon)$, it follows that $\text{TOPDOWN}_{\mathscr{G}}$ will make a split for which the true purity gain is indeed $\text{poly}(\varepsilon/t)$. By Fact 2.1, such a split constitutes good progress with respect to the potential function $\mathscr{G}\text{-impurity}_f$. Summarizing, [BLT20a] that shows until $\text{error}_f(T_S^\circ) < \text{opt}_s + \varepsilon$ is achieved, *every* split that $\text{TOPDOWN}_{\mathscr{G}}$ makes has high true purity gain, and hence constitutes good progress with respect to the potential function $\mathscr{G}\text{-impurity}_f$.

The key technical difficulty in analyzing $\text{MINIBATCHTOPDOWN}_{\mathscr{G}}$ instead of $\text{TOPDOWN}_{\mathscr{G}}$ is that $\text{MINIBATCHTOPDOWN}_{\mathscr{G}}$ is not guaranteed to choose a split with high true purity gain: it could make splits for which its estimate $\text{PurityGain}_{\mathscr{G},B}(\ell, i)$ is high, but the true purity gain $\text{PurityGain}_{\mathscr{G},f}(\ell, i)$ is actually tiny. In fact, unless we use batches of size $b \geq \text{poly}(t)$, exponentially larger than the $b = \text{polylog}(t)$ of Theorem 1, $\text{MINIBATCHTOPDOWN}_{\mathscr{G}}$ could make splits that result in zero true purity gain, and hence constitute zero progress with respect to the potential function $\mathscr{G}\text{-impurity}_f$.

To overcome this challenge, we instead show that *most* splits $\text{MINIBATCHTOPDOWN}_{\mathscr{G}}$ makes have high true purity gain. We first show that with high probability over the draws of minibatches $B$, if $\text{MINIBATCHTOPDOWN}_{\mathscr{G}}$ splits a leaf that is neither too shallow nor too deep within $T^\circ$, then this split has high true purity gain (Lemma 6.5). We then show the following two lemmas:

1. Lemma B.6: If $\text{MINIBATCHTOPDOWN}_{\mathscr{G}}$ splits a leaf of $T^\circ$ that is sufficiently deep, then it must be the case that $\text{error}_f(T^\circ_{\text{Batch}_b(S)}) \leq \text{opt}_s + \varepsilon$, i.e. the current tree already achieves sufficiently small error. With this Lemma, we are able to define $\text{MINIBATCHTOPDOWN}_{\mathscr{G}}$ to never split a leaf that is too deep, while retaining guarantees on its performance.

2. Lemma B.7: This lemma shows that only a small fraction of splits made by $\text{MINIBATCHTOPDOWN}_{\mathscr{G}}$ can be too shallow.

Combining the above Lemmas, we are able to prove Theorem 1. We defer the proof to Appendix 6.

## 3   Proof overviews for Theorems 2 and 3

We begin with a proof overview for Theorem 2. Let $T$ be the decision tree hypothesis that $\text{MINIBATCHTOPDOWN}_{\mathscr{G}}$ would construct if we were to all of $S^\circ$ and train $\text{MINIBATCHTOPDOWN}_{\mathscr{G}}$ on it. Our goal is to efficiently compute $T(x^\star)$ for a given $x^\star$ by selectively labeling only $q$ points within $S^\circ$, where $q$ is exponentially smaller than the sample complexity of learning and constructing $T$.

Intuitively, we would like $\text{LOCALLEARNER}_{\mathscr{G}}$ to only grow the single "strand" within $T$ required to compute $T(x^\star)$ instead of the entire tree $T$—this "strand" is simply the root-to-leaf path of $T$ that $x^\star$ follows. The key challenge that arises in implementing this plan is: how does $\text{LOCALLEARNER}_{\mathscr{G}}$ know when to terminate this strand (i.e. how does it know when it has reached a leaf of $T$)? $\text{MINIBATCHTOPDOWN}_{\mathscr{G}}$, the "global" algorithm that $\text{LOCALLEARNER}_{\mathscr{G}}$ is trying the simulate, terminates when the tree is of size $t$. As $\text{LOCALLEARNER}_{\mathscr{G}}$ grows the strand corresponding to $x^\star$, how could it estimate the size of the overall tree without actually growing it? In other words, it is not clear how one would define the stopping criterion of the while loop in the following pseudocode:

Roughly speaking, we want "*stopping criterion*" to answer the following question: if we grew a size-$t$ tree using $\text{MINIBATCHTOPDOWN}_{\mathscr{G}}$ (on the labeled version of $S^\circ$), would $\ell$ be a leaf of the resulting tree, or would it be an internal node? Nearly equivalently, with access to just a single strand of a tree, we wish to estimate the size of that tree. If that size is $t$, then we stop the while loop.

It is not possible to accurately estimate the size of a tree using just a single strand. However, by computing a small number of random strands, we can get an accurate size estimator. In Section 7, we show that for $x_1, \ldots, x_m$ chosen uniformly at random from $\{\pm 1\}^d$, the estimator $e := \frac{1}{m} \sum_{i=1}^{m} 2^{|\ell_T(x_i)|}$

Initialize $\ell$ to be the leaf of the empty tree.

while (*stopping criterion*) {

     1. Draw $\boldsymbol{B}^\circ \sim \mathrm{Batch}_b(S^\circ, \ell)$ and query $f$'s values on these points. Let $i^\star$ be the coordinate that maximizes $\mathrm{PurityGain}_{\mathscr{G}, \boldsymbol{B}^\circ}(\ell, i)$ among all $i \in [d]$.

     2. Extend $\ell$ according to the value of $x^\star_{i^\star}$.

}

Draw $\boldsymbol{B}^\circ \sim \mathrm{Batch}_b(S, \ell)$ and query $f$'s values on these points.

Output $\mathrm{round}(\mathbb{E}_{\boldsymbol{x} \sim \boldsymbol{B}^\circ}[f(\boldsymbol{x})])$.

accurately estimates the size of $T$, as long as the depth of $T$ is not too large. Therefore, rather than growing only the root-to-leaf path for $x^\star$, LOCALLEARNER$_{\mathscr{G}}$ samples random additional inputs, $\boldsymbol{x}_1, \ldots, \boldsymbol{x}_m$. Then, it simultaneously grows the strands for the root-to-leaf paths of $x^\star$ as well as $\boldsymbol{x}_1, \ldots, \boldsymbol{x}_m$. These strands do not all grow at the same "rate", as we want LOCALLEARNER$_{\mathscr{G}}$ to make splits in the same order as MINIBATCHTOPDOWN$_{\mathscr{G}}$ does. As long as it does this, we can use the size estimator to, at any step, accurately estimate the size of tree MINIBATCHTOPDOWN$_{\mathscr{G}}$ would need to build for all the current strands to end at leaves. LOCALLEARNER$_{\mathscr{G}}$ terminates when its estimate of this size is $t$.

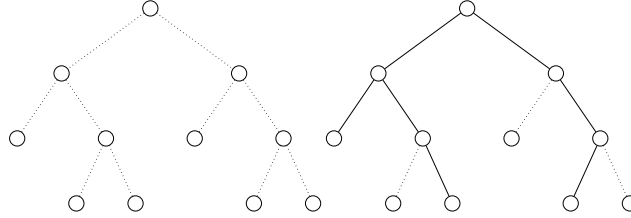

Figure 3: Rather than growing the entire tree $T$ (depcited on the LHS) as MINIBATCHTOPDOWN$_{\mathscr{G}}$ does, LOCALLEARNER$_{\mathscr{G}}$ only grows $m + 1$ strands within $T$ (depicted on the RHS), corresponding to the given input $x^\star$ and $m$ additional random inputs $\boldsymbol{x}_1, ..., \boldsymbol{x}_m \sim \{\pm 1\}^d$.

We back the above intuition for LOCALLEARNER$_{\mathscr{G}}$ with proofs. In Section 8, we show that the output of LOCALLEARNER$_{\mathscr{G}}$ for size parameter $t$ is $T(x^\star)$, where $T$ is size-$t'$ tree produced by MINIBATCHTOPDOWN$_{\mathscr{G}}$ where $t \in t'(1 \pm \eta)$. We also show that LOCALLEARNER$_{\mathscr{G}}$ needs to only label polylogarithmic many points within $S^\circ$ to compute $T(x^*)$. This completes our proof overview for Theorem 2, and Theorem 3 is a straightforward consequence of Theorem 2.

## 4 Conclusion

We have given strengthened provable guarantees on the performance of popular and empirically successful top-down decision tree learning heuristics such as ID3, C4.5, and CART, focusing on sample complexity. First, we designed and analyzed *minibatch* versions of these heuristics, MINIBATCHTOPDOWN$_{\mathscr{G}}$, and proved that they achieve the same performance guarantees as the full-batch versions. We then gave an implementation of MINIBATCHTOPDOWN$_{\mathscr{G}}$ within the recently-introduced active local learning framework of [BBG20]. Building on these results, we showed that MINIBATCHTOPDOWN$_{\mathscr{G}}$ is amenable to highly efficient learnability estimation [KV18, BH18]: its performance can be estimated accurately by selectively labeling very few examples.

As discussed in [KV18, BH18], this new notion of learnability estimation opens up a whole host of theoretical and empirical directions for future work. We discuss several concrete ones that are most relevant to our work. Our algorithm Est$_{\mathscr{G}}$ efficiently and accurately estimates the quality, relative to a test set $S_{\mathrm{test}}$, of the hypothesis that MINIBATCHTOPDOWN$_{\mathscr{G}}$ would produce if trained on a set $S^\circ$. Could Est$_{\mathscr{G}}$ be more broadly useful in assessing the quality of the *training data $S^\circ$ itself*, relative to $S_{\mathrm{test}}$? Could its estimates provide guarantees on the performance of other algorithms when trained in $S^\circ$ and tested on $S_{\mathrm{test}}$? It would also be interesting to explore applications of our algorithms to the design of training sets. Given training sets $S_1, \ldots, S_m$, Est$_{\mathscr{G}}$ allows us to efficiently determine

the $S_i$ for which MINIBATCHTOPDOWN$_{\mathscr{G}}$ would produce a hypothesis that achieves the smallest error with respect to $S_{\text{test}}$. Could Est$_{\mathscr{G}}$ or extensions of it be useful in efficiently *creating an $S^\star$*, comprising data from each $S_i$, that is of higher quality than any $S_i$ individually? Finally, while we have focused on top-down heuristics for learning a single decision tree in this work, a natural next step would be to design and analyze learnability estimation procedures for ensemble methods such as random forests and gradient boosted trees.

## Broader Impact

This work does not present any foreseeable societal consequence.

## Acknowledgements and Disclosure of Funding

Guy, Jane, and Li-Yang were supported by NSF award CCF-192179 and NSF CAREER award CCF-1942123. Neha was supported by NSF award 1704417 and Moses Charikar's Simons Investigator grant.

## Footnotes

[1] Impurity functions $\mathscr{G}$ are restricted to be concave, symmetric around $\frac{1}{2}$, and to satisfy $\mathscr{G}(0) = \mathscr{G}(1) = 0$ and $\mathscr{G}(\frac{1}{2}) = 1$. For example, ID3 and C4.5 use the binary entropy function $\mathscr{G}(p) = \mathrm{H}(p)$, and the associated purity gain is commonly referred to as information gain; CART uses the Gini criterion $\mathscr{G}(p) = 4p(1 - p)$; Kearns and Mansour proposed and analyzed the function $\mathscr{G}(p) = 2\sqrt{p(1 - p)}$ [KM99]. The work of Dietterich, Kearns, and Mansour [DKM96] provides a detailed discussion and experimental comparison of various impurity functions.

[2]To ensure that $\mathrm{LocalLearner}_\mathscr{G}$ consistently labels all $x^\star$ according to the *same* tree $T$, we run all invocations of $\mathrm{LocalLearner}_\mathscr{G}$ with the same outcomes of randomness for $\boldsymbol{B}^\circ_{\mathrm{strands}}$ and draws of minibatches.

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
