[Supplementary Material]

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

---

## B  Provable guarantees for MINIBATCHTOPDOWN

In this section we will prove Theorem 1. We first need a couple of definitions:

**Definition 3** (Hölder continuous). *For $C, \alpha > 0$, an impurity function $\mathscr{G} : [0,1] \to [0,1]$ is $(C, \alpha)$-Hölder continuous if, for all $a, b \in [0,1]$,*

$$|\mathscr{G}(a) - \mathscr{G}(b)| \le C|a - b|^{\alpha}.$$

**Definition 4** (Strong concavity). *For $\kappa > 0$, an impurity function $\mathscr{G} : [0,1] \to [0,1]$ is $\kappa$-strongly concave if for all $a, b \in [0,1]$,*

$$\frac{\mathscr{G}(a) + \mathscr{G}(b)}{2} \le \mathscr{G}\left(\frac{a+b}{2}\right) - \frac{\kappa}{2} \cdot (b-a)^2.$$

**Theorem 4** (Provable guarantee for MINIBATCHTOPDOWN; formal version of Theorem 1). *Let $f : \{\pm 1\}^d \to \{0, 1\}$ be a monotone target function and $\mathscr{G}$ be any $\kappa$-strongly concave and $(C, \alpha)$-Hölder continuous impurity function. For any $s \in \mathbb{N}$, $\varepsilon, \delta \in (0, \frac{1}{2})$, let $t = s^{\Theta(\log(s))/\varepsilon^2}$, and $\boldsymbol{S}$ be a set of $n$ labeled training examples $(\boldsymbol{x}, f(\boldsymbol{x}))$ where $\boldsymbol{x} \sim \{\pm 1\}^d$ is uniform random, and*

$$n = t \cdot \Omega\left( \left( \frac{C^2 \log(s)^4}{\kappa^2 \varepsilon^4} \right)^{\frac{1}{\alpha}} \cdot \log\left( \frac{td}{\delta} \right) \cdot \log t \right).$$

*If the minibatch size is at least*

$$b = \Omega\left( \left( \frac{C^2 \log(s)^4}{\kappa^2 \varepsilon^4} \right)^{\frac{1}{\alpha}} \cdot \log\left( \frac{td}{\delta} \right) \right),$$

*then with probability at least $1 - \delta$ over the randomness of $\boldsymbol{S}$ and the draws of minibatches from within $\boldsymbol{S}$, the size-$t$ decision tree hypothesis constructed by MINIBATCHTOPDOWN$_{\mathscr{G}}(t, b, \boldsymbol{S})$ satisfies $\text{error}_f(T) \le \text{opt}_s + \varepsilon$.*

### B.1  Properties of batches

We begin by specifying how large the batch size has to be for accurate estimates of local gain. Later on, we will turn accurate estimates of local gain to estimates of purity gain that are accurate at least half the time.

**Lemma B.1** (Every leaf has a batch of size $b_{min}$). *Let*

$$b_{\min} = \max \left( 8, 2 \cdot \left( \frac{2C}{\Delta} \right)^{\frac{2}{\alpha}} \right) \cdot \log_e \left( \frac{9td}{\delta} \right)$$

*Then with probability at least $1 - \frac{\delta}{3}$, every $\ell$ satisfying $|\ell| \leq \log(n/(2b_{\min}))$ that* MINIBATCHTOPDOWN$_{\mathscr{G}}(t, b, \boldsymbol{S})$ *constructs has a minibatch, $\boldsymbol{B} \sim \mathrm{Batch}_b(\boldsymbol{S}, \ell)$, of size at least $b_{\min}$.*

*Proof.* It suffices to show that the number of points in $\boldsymbol{S}$ consistent with each of these $\ell$ is at least $b_{\min}$. Fix any such $\ell$ satisfying $|\ell| \leq \log(n/(2b_{\min}))$. The probability an element in $\boldsymbol{S}$ is consistent with $\ell$ is at least $\frac{2b_{\min}}{n}$, meaning the expected number of points consistent with $\ell$ is at least $2b_{\min}$. By the multiplicative Chernoff bound,

$$\Pr \left[ \sum_{(x,y) \in S} \mathbb{1}[x_i \text{ consistent with } \ell] < b_{\min} \right] \leq \exp_e \left( -\frac{1}{8} \cdot 2b_{\min} \right)$$

There are at most $t$ leaves that MINIBATCHTOPDOWN$_{\mathscr{G}}(t, b, \boldsymbol{S})$ will ever estimate impurity gain for, so as long as,

$$b_{\min} \geq 4 \cdot \log_e \left( \frac{3t}{\delta} \right),$$

with probability at least $1 - \delta/3$, all of them will have a minibatch of size at least $b_{\min}$. $\qquad\square$

**Lemma B.2** (Batches are balanced). *With probability at least $1 - \delta/3$, there are at least $\frac{b_{\min}}{4}$ points $(x, y)$ in $\boldsymbol{B}$ satisfying $x_i = -1$ and $\frac{b_{\min}}{4}$ points satisfying $x_i = 1$.*

*Proof.* The mini batch $\boldsymbol{B}$ is formed by choosing at least $b_{\min}$ points that are consistent with $\ell$, without replacement, from $\boldsymbol{S}$, which is itself formed by taking points with replacement from $\{\pm 1\}^d$. This means that the mini batch $\boldsymbol{B}$ has at least $b_{\min}$ points without replacement from $\{\pm 1\}^d$. Fix any $\ell$ and let $b_{\mathrm{true}}$ be the number of points in $\boldsymbol{B}$. By Hoeffding's inequality,

$$\Pr \left[ \left| \frac{b_{\mathrm{true}}}{2} - (\text{Number of } (x, y) \in \boldsymbol{B} \text{ where } x_i = -1) \right| \geq \frac{b_{\mathrm{true}}}{4} \right] \leq \exp_e(-\frac{b_{\mathrm{true}}}{8})$$

$$\leq \exp_e(-\frac{b_{\min}}{8})$$

MINIBATCHTOPDOWN$_{\mathscr{G}}$ computes LocalGain$_{\mathscr{G}, \boldsymbol{B}}(\ell, i)$ for at most $t$ different $\ell$ and $d$ different $i$, for a total of $t \cdot d$ different computations. As long as

$$b_{\min} \geq 8 \cdot \log_e \left( \frac{3td}{\delta} \right),$$

then with probability at least $1 - \delta/3$, both $\boldsymbol{B}[x_i = -1]$ and $\boldsymbol{B}[x_i = 1]$ will have at least $\frac{b_{\mathrm{true}}}{4} \geq \frac{b_{\min}}{4}$ points. $\qquad\square$

**Lemma B.3** (Batch size is logarithmic in $td$). *For any $f : \{\pm 1\}^d \to \{0, 1\}$ and $n \in \mathbb{N}$, let $\boldsymbol{S}$ be a size $n$ sample of points $(\boldsymbol{x}, f(\boldsymbol{x}))$ where $\boldsymbol{x} \sim \{\pm 1\}^d$. Furthermore, let $\mathscr{G} : [0, 1] \to [0, 1]$ be any $(C, \alpha)$-Hölder continuous impurity function. For any $\Delta > 0$, and*

$$b \geq b_{\min} = \max \left( 8, 2 \cdot \left( \frac{2C}{\Delta} \right)^{\frac{2}{\alpha}} \right) \cdot \log_e \left( \frac{9td}{\delta} \right)$$

*with probability at least $1 - \delta$, any time* MINIBATCHTOPDOWN$_{\mathscr{G}}(t, b, \boldsymbol{S})$ *computes* LocalGain$_{\mathscr{G}, \boldsymbol{B}}(\ell, i)$ *for $|\ell| \leq \log(n/(2b_{\min}))$ for a mini batch $\boldsymbol{B} \sim \mathrm{Batch}_b(\boldsymbol{S}, \ell)$*

$$|\mathrm{LocalGain}_{\mathscr{G}, \boldsymbol{B}}(\ell, i) - \mathrm{LocalGain}_{\mathscr{G}, f}(\ell, i)| \leq \Delta$$

*Proof.* For any particular $\ell, i$, in order to compute $\text{LocalGain}_{\mathscr{G}, \boldsymbol{B}}$ we need to estimate three expectations:

$$\mathscr{G}(\mathbb{E}[f(\boldsymbol{x})]$$
$$\mathscr{G}(\mathbb{E}[f(\boldsymbol{x}) \mid \boldsymbol{x} \text{ reaches } \ell, \boldsymbol{x}_i = -1]$$
$$\mathscr{G}(\mathbb{E}[f(\boldsymbol{x}) \mid \boldsymbol{x} \text{ reaches } \ell, \boldsymbol{x}_i = 1]$$

Define $\varepsilon_1, \varepsilon_2, \varepsilon_3$ to be the errors made in computing these expectations so that

$$\text{LocalGain}_{\mathscr{G}, \boldsymbol{B}}(\ell, i) := \mathscr{G}(\mathbb{E}[f(\boldsymbol{x}) \mid \boldsymbol{x} \text{ reaches } \ell] + \varepsilon_1)$$
$$- \left(\tfrac{1}{2} \cdot \mathscr{G}(\mathbb{E}[f(\boldsymbol{x}) \mid \boldsymbol{x} \text{ reaches } \ell, \boldsymbol{x}_i = -1] + \varepsilon_2)\right.$$
$$+ \left.\tfrac{1}{2} \cdot \mathscr{G}(\mathbb{E}[f(\boldsymbol{x}) \mid \boldsymbol{x} \text{ reaches } \ell, \boldsymbol{x}_i = 1] + \varepsilon_3)\right).$$

Suppose that $\varepsilon_1, \varepsilon_2, \varepsilon_3$ are each bounded as

$$|\varepsilon_j| \le \left(\frac{\Delta}{2C}\right)^{\frac{1}{\alpha}} \tag{1}$$

Then, by the definition of Hölder continuous and triangle inequality,

$$|\text{LocalGain}_{\mathscr{G}, \boldsymbol{B}}(\ell, i) - \text{LocalGain}_{\mathscr{G}, f}(\ell, i)|$$
$$\le |\mathscr{G}(\mathbb{E}[f(\boldsymbol{x}) \mid \boldsymbol{x} \text{ reaches } \ell] + \varepsilon_1) - \mathscr{G}(\mathbb{E}[f(\boldsymbol{x}) \mid \boldsymbol{x} \text{ reaches } \ell])|$$
$$+ \frac{1}{2}|\mathscr{G}(\mathbb{E}[f(\boldsymbol{x}) \mid \boldsymbol{x} \text{ reaches } \ell, \boldsymbol{x}_i = -1] + \varepsilon_2) - \mathscr{G}(\mathbb{E}[f(\boldsymbol{x}) \mid \boldsymbol{x} \text{ reaches } \ell, \boldsymbol{x}_i = -1])|$$
$$+ \frac{1}{2}|\mathscr{G}(\mathbb{E}[f(\boldsymbol{x}) \mid \boldsymbol{x} \text{ reaches } \ell, \boldsymbol{x}_i = 1] + \varepsilon_3) - \mathscr{G}(\mathbb{E}[f(\boldsymbol{x}) \mid \boldsymbol{x} \text{ reaches } \ell, \boldsymbol{x}_i = 1])|$$
$$\le C \cdot (|\varepsilon_1|^\alpha + \frac{1}{2} \cdot |\varepsilon_2|^\alpha + \frac{1}{2} \cdot |\varepsilon_3|^\alpha)$$
$$\le \Delta$$

Therefore, it is enough to show that for all $\ell, i$, the corresponding $\varepsilon_1, \varepsilon_2, \varepsilon_3$ satisfy Equation (1).

By Lemma B.1 and Lemma B.2, with high probability all of these expectations are over at least $\frac{b_{\min}}{4}$ terms. Given the above is true, we can use Hoeffding's inequality to bound each $\varepsilon_j$,

$$\Pr\left[|\varepsilon_j| > \left(\frac{\Delta}{2C}\right)^{\frac{1}{\alpha}}\right] \le \exp_e\left(-2 \cdot \frac{b_{\min}}{4} \cdot \left(\frac{\Delta}{2C}\right)^{\frac{2}{\alpha}}\right).$$

There are a total of at most $3td$ such $\varepsilon_j$ we wish to bound. Setting $b_{\min}$ to at least

$$2 \cdot \left(\frac{2C}{\Delta}\right)^{\frac{2}{\alpha}} \cdot \log_e\left(\frac{9td}{\delta}\right)$$

means all are bounded as desired with probability at least $1 - \delta/3$. $\qquad\square$

## B.2 Properties of MiniBatchTopDown

As we discussed in Section 2, a key component of [BLT20a]'s analysis is a proof that if $\text{error}_f(T_S^\circ) > \text{opt}_s + \varepsilon$, there must exist a leaf $\ell^\star \in T^\circ$ and a coordinate $i^\star \in [d]$ such that

$$\text{PurityGain}_{\mathscr{G}, f}(\ell^\star, i^\star) > \frac{\kappa\varepsilon^2}{32j(\log s)^2}. \tag{2}$$

Based on how we set $\Delta$ in Lemma B.3, MINIBATCHTOPDOWN$_{\mathscr{G}}$ will be able to estimate all local gains to additive accuracy $\pm O(\frac{\kappa\varepsilon^2}{\log(s)^2})$. That accuracy, in conjunction with just Equation (2), is *not* sufficient to prove that MINIBATCHTOPDOWN$_{\mathscr{G}}$ will produce a low error tree. Instead, we need the following additional fact that [BLT20a] proved one step prior to showing Equation (2); in fact, it implies Equation (2) but is stronger, and that strength is needed for our purposes.

**Fact B.4** (Showed during the proof of Theorem 2 of [BLT20a])**.** *Let $T^\circ$ be any partial tree. For any $f : \{\pm 1\}^d \to \{0,1\}$ and $\kappa$-strongly concave impurity function $\mathscr{G} : [0,1] \to [0,1]$, if $\mathrm{error}_f(T_S^\circ) > \mathsf{opt}_s + \varepsilon$, then*

$$\sum_{\text{leaves } \ell \,\in\, T^\circ} \max_{i \in [d]} \left( \mathrm{PurityGain}_{\mathscr{G},f}(\ell, i) \right) > \frac{\kappa}{32} \cdot \left( \frac{\varepsilon}{\log s} \right)^2.$$

Fact B.4 implies Equation (2) because, if $T^\circ$ is size $j$ and the total purity gains of all of its leaves is some value $z$, then at least one leaf has purity gain $\frac{z}{j}$. We use Fact B.4 to show that, whenever $\textsc{MiniBatchTopDown}_\mathscr{G}$ picks a leaf that is neither too deep nor too high in the tree, it has picked a leaf and index with relatively large purity gain.

**Lemma B.5** (Medium depth splits are good.)**.** *Choose any max depth $D \in \mathbb{N}$. Let $f : \{\pm 1\}^d \to \{0,1\}$ be a monotone target function and $\mathscr{G}$ be any $\kappa$-strongly concave and $(C, \alpha)$-Hölder continuous impurity function. For any $s \in \mathbb{N}$, $\varepsilon, \delta \in (0, \frac{1}{2})$, let $t = s^{O(\log(s))/\varepsilon^2}$, and $S$ be a set of $n$ labeled examples $(x, f(x))$ where $x \sim \{\pm 1\}^d$ is uniform random,*

$$n = \Omega \left( \left( \frac{C^2 \log(s)^4}{\kappa^2 \varepsilon^4} \right)^{\frac{1}{\alpha}} \cdot \log\left( \frac{td}{\delta} \right) \cdot 2^D \right)$$

*and*

$$b = \Omega \left( \left( \frac{C^2 \log(s)^4}{\kappa^2 \varepsilon^4} \right)^{\frac{1}{\alpha}} \cdot \log\left( \frac{td}{\delta} \right) \right).$$

*Then, with probability at least $1 - \delta$, the following holds for all iterations of $\textsc{MiniBatchTopDown}_\mathscr{G}(t, b, S)$. If, at iteration $j$, $T^\circ$ satisfies,*

$$\mathrm{error}_f(T^\circ_{\mathrm{Batch}_b(S)}) \geq \mathsf{opt}_s + 2\varepsilon,$$

*let $(\ell^\star, i^\star)$ be the leaf and coordinate chosen to maximize the $\mathrm{PurityGain}_{\mathscr{G},B}$. Then, if*

$$\log(j) - 2 \leq |\ell^\star| \leq D,$$

*then*

$$\mathrm{PurityGain}_{\mathscr{G},f}(\ell^\star, i^\star) > \frac{\kappa}{64} \cdot \frac{\varepsilon^2}{j(\log s)^2}.$$

*Proof.* For the values of $n$ and $b$ given in this lemma statement, using Lemma B.3, we have for $\Delta = \frac{\kappa}{32 \cdot 10} \cdot \left( \frac{\varepsilon}{\log s} \right)^2$, for all leaves with $|l| \leq D$

$$|\mathrm{LocalGain}_{\mathscr{G},B}(\ell, i) - \mathrm{LocalGain}_{\mathscr{G},f}(\ell, i)| \leq \frac{\kappa}{32 \cdot 10} \cdot \left( \frac{\varepsilon}{\log s} \right)^2 \tag{3}$$

with probability atleast $1 - \delta$.

Since $\mathrm{error}_f((T^\circ)_{\mathrm{Batch}_b(S)}) \geq \mathsf{opt}_s + 2\varepsilon$ and using lemma B.1, we know that $\mathrm{error}_f((T^\circ)_S) \geq \mathsf{opt}_s + \varepsilon$ since the batch size is large enough, we can use Fact B.4 to lower bound the estimated purity gain of $\ell^\star$ and $i^\star$. Let $c = \frac{\kappa}{32}$

$$\sum_{\text{leaves } \ell \,\in\, T^\circ} \max_{i \in [d]} \left( \mathrm{PurityGain}_{\mathscr{G},f}(\ell, i) \right) > c \cdot \left( \frac{\varepsilon}{\log s} \right)^2 \qquad \text{Fact B.4}$$

$$\sum_{\text{leaves } \ell \,\in\, T^\circ} 2^{-|\ell|} \cdot \max_{i \in [d]} \left( \mathrm{LocalGain}_{\mathscr{G},f}(\ell, i) \right) > c \cdot \left( \frac{\varepsilon}{\log s} \right)^2$$

$$\sum_{\text{leaves } \ell \,\in\, T^\circ} 2^{-|\ell|} \cdot \max_{i \in [d]} \left( \mathrm{LocalGain}_{\mathscr{G},B}(\ell, i) \right) > \frac{9c}{10} \cdot \left( \frac{\varepsilon}{\log s} \right)^2 \qquad \text{Equation (3) and } \sum_\ell 2^{-|\ell|} = 1$$

$$\sum_{\text{leaves } \ell \,\in\, T^\circ} \max_{i \in [d]} \left( \mathrm{PurityGain}_{\mathscr{G},B}(\ell, i) \right) > \frac{9c}{10} \cdot \left( \frac{\varepsilon}{\log s} \right)^2$$

417 Since there are $j$ leaves in $T^\circ$ and $\ell^\star, i^\star$ are chosen to maximize $\mathrm{PurityGain}_{\mathscr{G},\boldsymbol{B}}(\ell^\star, i^\star)$,

$$\mathrm{PurityGain}_{\mathscr{G},\boldsymbol{B}}(\ell^\star, i^\star) > \frac{9c}{10j} \cdot \left(\frac{\varepsilon}{\log s}\right)^2.$$

418 Next, we show that since $\ell^*$ is sufficiently far down in the tree, then the estimated purity gain and
419 true purity gain are close.

$$|\mathrm{PurityGain}_{\mathscr{G},\boldsymbol{B}}(\ell^\star, i^\star) - \mathrm{PurityGain}_{\mathscr{G},f}(\ell^\star, i^\star)|$$
$$= 2^{-|\ell^\star|} \cdot |\mathrm{LocalGain}_{\mathscr{G},\boldsymbol{B}}(\ell^\star, i^\star) - \mathrm{LocalGain}_{\mathscr{G},f}(\ell^\star, i^\star)|$$
$$\leq 2^{-|\ell^\star|} \cdot \frac{c}{10} \cdot \left(\frac{\varepsilon}{\log s}\right)^2 \qquad\qquad \text{eq. (3)}$$
$$\leq \frac{4}{j} \cdot \frac{c}{10} \cdot \left(\frac{\varepsilon}{\log s}\right)^2 \qquad\qquad |\ell^\star| \geq \log(j) - 2$$

420 By triangle inequality, we have that $\mathrm{PurityGain}_{\mathscr{G},f}(\ell^\star, i^\star) > \frac{c}{2j} \cdot \left(\frac{\varepsilon}{\log s}\right)^2$, the desired result.

421 $\qquad\qquad\qquad\qquad\qquad\qquad\qquad\qquad\qquad\qquad\qquad\qquad\qquad\qquad\qquad\qquad\qquad\qquad\quad \square$

422 Given that we are only guaranteed to make good progress on splits that are neither too deep nor too
423 shallow, we will need to deal with both possibilities. First, we show that if we ever wanted to make
424 too deep a split, we would already be done.

425 **Lemma B.6** (Can stop at very large depth.)**.** *Let $f : \{\pm 1\}^d \to \{0, 1\}$ be a monotone target function*
426 *and $\mathscr{G}$ be any $\kappa$-strongly concave and $(C, \alpha)$-Hölder continuous impurity function. For any $s \in \mathbb{N}$,*
427 *$\varepsilon, \delta \in (0, \frac{1}{2})$, let*

$$t = s^{\Theta(\log(s))/(\kappa \varepsilon^2)}, \qquad\qquad\qquad (4)$$

428 *set the max depth to*

$$D = \lfloor \log(t) + \log\log t \rfloor, \qquad\qquad\qquad (5)$$

429 *let $\boldsymbol{S}$ be a set of $n$ labeled examples $(\boldsymbol{x}, f(\boldsymbol{x}))$ where $\boldsymbol{x} \sim \{\pm 1\}^d$ is uniform random,*

$$n = \Omega\left(\left(\frac{C^2 \log(s)^4}{\kappa^2 \varepsilon^4}\right)^{\frac{1}{\alpha}} \log\left(\frac{td}{\delta}\right) \cdot 2^D\right) = \mathrm{poly}_{\alpha,\kappa,C}(t, \log(d), \log(1/\delta)),$$

430 *and batch size at least*

$$b = \Omega\left(\left(\frac{C^2 \log(s)^4}{\kappa^2 \varepsilon^4}\right)^{\frac{1}{\alpha}} \log\left(\frac{td}{\delta}\right)\right).$$

431 *Let $T_1^\circ, T_2^\circ, \ldots, T_t^\circ$ be the size $1, 2, \ldots, t$ partials trees that $\mathrm{MINIBATCHTOPDOWN}_{\mathscr{G}}(t, b, \boldsymbol{S})$ builds.*
432 *With probability $1 - \delta$ over the randomness of $\boldsymbol{S}$ and the random batches, for any $k \in [t]$, if $T_k^\circ$ has*
433 *depth more than $D$, then*

$$\mathrm{error}_f((T_k^\circ)_{\mathrm{Batch}_b(S)}) \leq \mathsf{opt}_s + 2\varepsilon. \qquad\qquad\qquad (6)$$

434 *Proof.* Let $k$ be chosen so that $T_k^\circ$ has depth more than $D$. For some $j \leq k$, there was a leaf $\ell^\star \in T_j^\circ$
435 that was split, satisfying,

$$|\ell^\star| = \lfloor \log(t) + \log\log t \rfloor - 1$$
$$= \lfloor \log(t) + \log\left(\Theta\left((\log s)^2/(\kappa\varepsilon^2)\right)\right) \rfloor - 1.$$

436 For any $i \in [d]$,

$$\mathrm{PurityGain}_{\mathscr{G},f}(\ell^\star, i) = 2^{-|\ell^\star|} \mathrm{LocalGain}_{\mathscr{G},f}(\ell^\star, i)$$
$$\leq 2^{-|\ell^\star|}$$
$$\leq \frac{1}{t} \cdot \Theta\left(\left(\frac{(\log s)^2}{\kappa\varepsilon^2}\right)^{-1}\right)$$
$$\leq \frac{1}{j} \cdot \Theta\left(\frac{\kappa\varepsilon^2}{(\log s)^2}\right). \qquad\qquad\qquad (7)$$

Note that the constant in Equation (7) is inversely related to the constant in the exponent of Equation (4). In Lemma B.5, we showed that if $\mathrm{error}_f((T_j^\circ)_{\mathrm{Batch}_b(S)}) \geq \mathsf{opt}_s + \varepsilon$, then for some $i^\star \in [d]$,

$$\mathrm{PurityGain}_{\mathscr{G},f}(\ell^\star, i^\star) = \Omega\left(\frac{\kappa \varepsilon^2}{j(\log s)^2}\right). \tag{8}$$

If we choose the constant in Equation (7) sufficiently low, which can be done by making the constant in Equation (4) sufficiently high, then that equation can not be satisfied at the same time as Equation (8). Therefore, it must be that $\mathrm{error}_f((T_j^\circ)_{\mathrm{Batch}_b(S)}) < \mathsf{opt}_s + \varepsilon$. Since $j < k$, and adding splits can only increase error by atmost $\varepsilon$, it must also be the case that $\mathrm{error}_f((T_k^\circ)_{\mathrm{Batch}_b(S)}) < \mathsf{opt}_s + 2\varepsilon$. $\qquad \square$

We next show that before Lemma B.6 kicks in, most splits are sufficiently deep to make good progress.

**Lemma B.7** (Few splits are shallow)**.** *Let $k = 2^a$ be any power of $2$ and $T_1^\circ, \ldots, T_k^\circ$ be a series of bare trees of size $1, \ldots, k$ respectively where $T_{j+1}$ is formed by splitting $\ell_j \in T_j$. Then,*

$$\sum_{j=1}^{k} \mathbb{1}\big[\, |\ell_j| < \log(j) - 2 \,\big] \leq \frac{k}{4}$$

*Proof.* First, since for all $j = 1, \ldots, k$, $j \leq k$, we can bound,

$$\sum_{j=1}^{k} \mathbb{1}\big[\, |\ell_j| < \log(j) - 2 \,\big] \leq \sum_{j=1}^{k} \mathbb{1}\big[\, |\ell_j| < \log(k) - 2 \,\big] = \sum_{j=1}^{k} \mathbb{1}\big[\, |\ell_j| < a - 2 \,\big].$$

If $\ell_j$, a leaf of $T_j^\circ$, has depth less than $a - 2$, then it is also an internal node of $T_k^\circ$ with depth less than $a - 2$. There are at most $2^{a-2} - 1$ nodes in any tree of depth less than $a - 2$. Therefore,

$$\sum_{j=1}^{k} \mathbb{1}\big[\, |\ell_j| \leq a - 2 \,\big] \leq 2^{a-2} - 1 \leq \frac{k}{4} - 1 \leq \frac{k}{4}.$$

$\qquad\qquad\qquad\qquad\qquad\qquad\qquad\qquad\qquad\qquad\qquad\qquad\qquad\qquad\qquad\qquad\qquad\qquad\qquad\quad\square$

## B.3 Final proof of Theorem 1

*Proof.* $\mathrm{MINIBATCHTOPDOWN}_\mathscr{G}$ builds a series of bare trees, $T_1^\circ, T_2^\circ, \ldots, T_t^\circ$, where $T_j$ has size $j$. We wish to prove that $\mathrm{error}_f((T_t^\circ)_{\mathrm{Batch}_b(S)}) \leq \mathsf{opt}_s + 3\varepsilon$ (In the end, we can choose $\varepsilon$ appropriately to get error $\mathsf{opt}_s + \varepsilon$). To do so, we consider two cases.

**Case 1**: There is some $k < t$ for which $\mathrm{error}_f((T_k^\circ)_{\mathrm{Batch}_b(S)}) \leq \mathsf{opt}_s + 2\varepsilon$.
Since splitting more variables of $T_k$ can only increase it's error by at most $\varepsilon$,

$$\mathrm{error}_f((T_t^\circ)_{\mathrm{Batch}_b(S)}) \leq \mathrm{error}_f((T_k^\circ)_{\mathrm{Batch}_b(S)}) + \varepsilon \leq \mathsf{opt}_s + 3\varepsilon,$$

which is the desired result.

**Case 2**: There is no $k < t$ for which $\mathrm{error}_f((T_k^\circ)_{\mathrm{Batch}_b(S)}) \leq \mathsf{opt}_s + \varepsilon$.

In this case, we use Lemma B.5 to ensure we make good progress. Lemma B.5 only applies when the tree has depth at most $D = \log t + \log \log t$. Luckily, Lemma B.6 ensures that if the tree has depth more than $D$, then we are ensured that $\mathrm{error}_f((T_t^\circ)_{\mathrm{Batch}_b(S)}) \leq \mathsf{opt}_s + 2\varepsilon$, and so are done. For the remainder of this proof, we assume all partial trees have depth at most $D$.

We will show that $\mathscr{G}\text{-impurity}(T_t^\circ) = 0$, which means that $\mathrm{error}_f((T_t^\circ)_{\mathrm{Batch}_b(S)}) = 0 \leq \mathsf{opt}_s + 2\varepsilon$, also proving the desired result. For $j = 1, \ldots, t-1$, let $\ell_j$ be the leaf of $T_j$ that is split, and $i_j$ be the coordinate placed at $\ell_j$ to form $T_{j+1}$. Then,

$$\mathscr{G}\text{-impurity}(T_t^\circ) = \mathscr{G}\text{-impurity}(T_1^\circ) - \sum_{j=1}^{t-1} \mathrm{PurityGain}_{\mathscr{G},f}(\ell_j, i_j).$$

467  Since $\mathscr{G}$-impurity$(T_1^\circ) \leq 1$ and our goal is to show that $\mathscr{G}$-impurity$(T_{t+1}^\circ) = 0$, it is sufficient to

468  show that $\sum_{j=1}^t \text{PurityGain}_{\mathscr{G},f}(\ell_j, i_j) \geq 1$. Lemma B.5 combined with Fact B.4,

$$\sum_{j=1}^t \text{PurityGain}_{\mathscr{G},f}(\ell_j, i_j) \geq \sum_{j=1}^t \mathbb{1}\left[\,|\ell_j| \geq \log(j) - 2\,\right] \cdot \frac{\kappa}{64j} \cdot \left(\frac{\varepsilon}{\log s}\right)^2$$

469  We break the above summation into chunks from $j = (2^a + 1)$ to $j = 2^{a+1}$, integer $a \leq \log(t)$. In

470  such a chunk, there are $2^a$ choices for $j$. By Lemma B.7, we know that for at most $2^{a+1}/4 = 2^a/2$

471  of those $j$ is $\mathbb{1}\left[\,|\ell_j| < \log(j) - 2\,\right]$. Therefore,

$$\sum_{j=2^a+1}^{2^{a+1}} \mathbb{1}\left[\,|\ell_j| \leq \log(j) - 2\,\right] \cdot \frac{\kappa}{64j} \cdot \left(\frac{\varepsilon}{\log s}\right)^2 \geq \frac{2^a}{2} \cdot \frac{\kappa}{64 \cdot (2^{a+1})} \cdot \left(\frac{\varepsilon}{\log s}\right)^2$$

$$= \frac{\kappa}{256} \cdot \left(\frac{\varepsilon}{\log s}\right)^2$$

472  Summing up $\frac{256}{\kappa} \cdot \left(\frac{\log s}{\varepsilon}\right)^2$ such chunks gives a sum of at least 1. Therefore, for

$$t = \exp\left(\Omega\left(\frac{(\log s)^2}{\kappa \varepsilon^2}\right)\right)$$

473  it must be the case that $\mathscr{G}$-impurity$(T_{t+1}^\circ) = 0$, proving the desired result.  $\square$

## C  Estimating the size of a decision tree

475  In this section, we design a decision tree size estimator. This size estimator only needs to inspect a

476  small number of random strands from the decision tree. It is unbiased, and as long as the decision

477  tree has a bounded max depth, obeys concentration bounds shown in Lemma C.1.

478  **Lemma C.1** (Size estimator). *For any $\Delta, \delta > 0$ and size-$s$ decision tree $T$, let $\ell^\star$ be the deepest leaf*

479  *in $T$ and*

$$m = \frac{(2^{|\ell^\star|})^2}{2\Delta^2} \cdot \ln\left(\frac{2}{\delta}\right).$$

480  *Choose $\boldsymbol{x}_1, \ldots, \boldsymbol{x}_m$ uniformly random from $\{\pm 1\}^d$ and define the estimator*

$$e := \frac{1}{m} \sum_{i=1}^m 2^{|\ell_T(\boldsymbol{x}_i)|}.$$

481  *With probability at least $1 - \delta$,*

$$|e - s| \leq \Delta.$$

482  *Proof.* We first show that $\mathbb{E}[e] = s$.

$$\mathbb{E}[e] = \mathbb{E}_{\boldsymbol{x} \sim \{\pm 1\}^d}\left[2^{|\ell_T(\boldsymbol{x})|}\right]$$

$$= \sum_{\text{leaves } \ell \in T} \Pr[\boldsymbol{x} \text{ reaches } \ell] \cdot 2^{|\ell|}$$

$$= \sum_{\text{leaves } \ell \in T} \frac{1}{2^{|\ell|}} \cdot 2^{|\ell|}$$

$$= s,$$

483  where the last equality is due to the fact that a size-$s$ tree has $s$ leaves. Furthermore, $e$ is the sum of

484  $m$ independent random variables bounded between 0 and $2^{|\ell^\star|}$. Therefore, we can apply Hoeffding's

485  inequality,

$$\Pr[|e - s| \geq \Delta] \leq 2\exp_e\left(-\frac{2m\Delta^2}{(2^{|\ell^\star|})^2}\right).$$

486  Plugging in $m$ proves the desired result.  $\square$

## D Provable guarantees for LOCALLEARNER

In order to facilitate comparisons between the output of LOCALLEARNER$_\mathscr{G}$ and MINIBATCHTOPDOWN$_\mathscr{G}$, we will define another algorithm, TOPDOWNSIZEESTIMATE$_\mathscr{G}$, that shares some elements with LOCALLEARNER$_\mathscr{G}$ and some elements with MINIBATCHTOPDOWN$_\mathscr{G}$.

---

TOPDOWNSIZEESTIMATE$_\mathscr{G}(t, b, S)$:

Initialize $T^\circ$ to be the empty tree.

Define $D \coloneqq \log t + \log \log t$.

Let $\boldsymbol{B}^\circ_{\mathrm{strands}}$ be $b$ uniform random points from $\{\pm 1\}^d$.

Initialize $e \coloneqq 1$, our size estimate.

while $(e < t)$ {

    1. *Score:* For each leaf $\ell \in T^\circ$ of depth at most $D$, draw $\boldsymbol{B} \sim \mathrm{Batch}_b(S, \ell)$. For each coordinate $i \in [d]$, compute:

$$\mathrm{PurityGain}_{\mathscr{G}, \boldsymbol{B}}(\ell, i) \coloneqq 2^{-|\ell|} \cdot \mathrm{LocalGain}_{\mathscr{G}, \boldsymbol{B}}(\ell, i), \text{ where}$$
$$\mathrm{LocalGain}_{\mathscr{G}, \boldsymbol{B}}(\ell, i) \coloneqq \mathscr{G}(\mathbb{E}[f(\boldsymbol{x})])$$
$$- \left( \tfrac{1}{2}\mathscr{G}(\mathbb{E}[\, f(\boldsymbol{x}) \mid \boldsymbol{x}_i = -1]) + \tfrac{1}{2}\mathscr{G}(\mathbb{E}[\, f(\boldsymbol{x}) \mid \boldsymbol{x}_i = 1]) \right),$$

    where the expectations are with respect to $(\boldsymbol{x}, f(\boldsymbol{x})) \sim \boldsymbol{B}$.

    2. *Split:* Let $(\ell^\star, i^\star)$ be the tuple that maximizes $\mathrm{PurityGain}_{\mathscr{G}, \boldsymbol{B}}(\ell, i)$. Grow $T^\circ$ by splitting $\ell^\star$ with a query to $x_{i^\star}$.

    3. *Estimate size:* Update our size estimate to

$$e = \mathop{\mathbb{E}}_{\boldsymbol{x} \in X}[2^{|\ell_{T^\circ}(\boldsymbol{x})|}]$$

}

For each leaf $\ell \in T^\circ$, draw $\boldsymbol{B} \sim \mathrm{Batch}_b(S, \ell)$ and label $\ell$ with $\mathrm{round}(\mathbb{E}_{(\boldsymbol{x}, f(\boldsymbol{x})) \sim \boldsymbol{B}}[f(\boldsymbol{x})])$.

---

Figure 4: TOPDOWNSIZEESTIMATE$_\mathscr{G}$ takes as input a size parameter $t$, a minibatch size $b$, and a labeled dataset $S$. It outputs a size-$t'$ decision tree hypothesis for $f$, where $t'$ is close to $t$.

**Comparison between MINIBATCHTOPDOWN$_\mathscr{G}$ and TOPDOWNSIZEESTIMATE$_\mathscr{G}$:** The only difference between MINIBATCHTOPDOWN$_\mathscr{G}$ and TOPDOWNSIZEESTIMATE$_\mathscr{G}$ is the stopping criterion. MINIBATCHTOPDOWN$_\mathscr{G}$ stops when the size of $T^\circ$ is exactly $t$. On the other hand, TOPDOWNSIZEESTIMATE$_\mathscr{G}$ estimates the size of $T^\circ$ using the estimator from Appendix C and stops when this size estimate is at least $t$.

**Comparison between LOCALLEARNER$_\mathscr{G}$ and MINIBATCHTOPDOWN$_\mathscr{G}$:** For any $t, S, b, x^\star$ that are valid inputs to LOCALLEARNER$_\mathscr{G}$, we compare the following two procedures.

    1. Running TOPDOWNSIZEESTIMATE$_\mathscr{G}(t, b, S)$ to get a decision tree, $T$, and then computing $T(x^\star)$.

    2. Only running LOCALLEARNER$_\mathscr{G}(t, b, S, x^\star)$.

We claim the output from the above two procedures is identical (given Footnote 2). TOPDOWNSIZEESTIMATE$_\mathscr{G}$ expands all paths in the tree its building, whereas LOCALLEARNER$_\mathscr{G}$ only expands paths that are pertinent to either the input $x^\star$, or inputs in $\boldsymbol{B}^\circ_{\mathrm{strands}}$, which are used to compute the size estimate. Aside from that, both of the above procedures are identical. Furthermore, paths not containing $x^\star$ nor any inputs in $\boldsymbol{B}^\circ_{\mathrm{strands}}$ have no effect on how the tree eventually labels $x^\star$. Therefore, the output of the two above procedures is identical, though LOCALLEARNER$_\mathscr{G}$ is more efficient as it only computes necessary paths.

Combining the above observations, we are able to prove the formal version of Theorem 2.

509 **Theorem 5** (Formal version of Theorem 2). *Let $f : \{\pm 1\}^d \to \{0,1\}$ be a target function, $\mathscr{G}$ be an*
510 *impurity function, and $S^\circ$ be an unlabeled training set.*

511 *For all $t \in \mathbb{N}$ and $\eta, \delta \in (0, \frac{1}{2})$, if the minibatch size is at least*

$$b = \Omega\left(\frac{(\log t)^2}{\eta^2} \cdot \log\left(\frac{t}{\delta}\right)\right),$$

512 *then with probability at least $1 - \delta$ over the randomness of $\boldsymbol{B}^\circ_{\mathrm{strands}}$, there is some $t' \in [t - \eta t, t + \eta t]$*
513 *for which the following holds. For all $x^\star \in \{\pm 1\}^d$, $\mathrm{LOCALLEARNER}_{\mathscr{G}}(t, b, S^\circ, x^\star)$ labels*

$$q = O(b^2 \log t)$$

514 *points within $S^\circ$ and returns $T(x^\star)$, where $T$ is the size-$t'$ decision tree hypothesis that*
515 *$\mathrm{MINIBATCHTOPDOWN}_{\mathscr{G}}(t', b, S)$ would construct, and $S$ is the labeled dataset obtained by la-*
516 *beling all of $S^\circ$ with $f$'s values.*

517 We break the proof of Theorem 5 into two pieces. First, we show that it labels only $O(b^2 \log t)$ points
518 within $S^\circ$, and then the rest.

519 **Lemma D.1** (Label efficiency of $\mathrm{LOCALLEARNER}_{\mathscr{G}}$). *Let $f : \{\pm 1\}^d \to \{0,1\}$ be a target function,*
520 *$\mathscr{G}$ be an impurity function, and $S^\circ$ be an unlabeled training set. For any $b, t \in \mathbb{N}$ and $x^\star \in \{\pm 1\}^d$,*
521 *$\mathrm{LOCALLEARNER}_{\mathscr{G}}(t, b, S^\circ, x^\star)$ labels at most*

$$q = O(b^2 \log t)$$

522 *points within $S^\circ$.*

523 *Proof.* It is sufficient for us to show that $\mathrm{LOCALLEARNER}_{\mathscr{G}}$ labels at most $O(b \log t)$ batches.
524 $\mathrm{LOCALLEARNER}_{\mathscr{G}}$ builds a series of bare trees $T_1^\circ, \ldots, T_{t'}^\circ$. During the while loop, the number of
525 batches it labels is equal to nodes in the following set

$$L := \bigcup_{j=1}^{t'} \left\{ \ell_{T_j^\circ}(x) \colon x \in \boldsymbol{B}^\circ_{\mathrm{strands}} \cup \{x^\star\}, |\ell_{T_j^\circ}(x)| \le D \right\}$$

526 Consider a single $x \in \boldsymbol{B}^\circ_{\mathrm{strands}} \cup \{x^\star\}$, and define

$$L(x) := \left\{ \ell_{T_j^\circ}(x) \colon j \in [t'], |\ell_{T_j^\circ}(x)| \le D \right\}.$$

527 Every node in $L(x)$ has depth at most $D$, and there is at most one node in $L(x)$ per depth. Therefore,
528 $|L(x)| \le D$, and

$$\begin{aligned}
|L| &\le \sum_{x \in \boldsymbol{B}^\circ_{\mathrm{strands}} \cup \{x^\star\}} |L(x)| \\
&\le (b+1)D \\
&= O(b \log t).
\end{aligned}$$

529 Therefore, $\mathrm{LOCALLEARNER}_{\mathscr{G}}$ labels only $O(b \log t)$ batches during the while loop. After the while
530 loop, it labels at most 1 additional batches. Therefore, it labels a total of $O(b \log t)$ batches which
531 requires labeling $O(b^2 \log t)$ points. $\qquad\square$

532 We next prove the remainder of Theorem 5.

533 *Proof.* Let $T$ be the tree that $\mathrm{TOPDOWNSIZEESTIMATE}_{\mathscr{G}}(t, b, S)$ produces. In the comparison
534 between $\mathrm{LOCALLEARNER}_{\mathscr{G}}$ and $\mathrm{TOPDOWNSIZEESTIMATE}_{\mathscr{G}}$, we established that, for all $x^\star \in$
535 $\{\pm 1\}^d$,

$$\mathrm{LOCALLEARNER}_{\mathscr{G}}(t, b, S, x^\star) = T(x^\star).$$

536 Set $t' = |T|$. Then, $T$ is also the output of $\mathrm{MINIBATCHTOPDOWN}_{\mathscr{G}}(t', b, \boldsymbol{S})$, as desired. Next, we
537 prove that $t' \in [t - \eta t, t + \eta t]$ with probability at least $1 - \delta$.

Let $T_1^\circ, T_2^\circ, \ldots, T_{t'}^\circ$ be the bare trees of size $1, 2, \ldots, t'$ that TOPDOWNSIZEESTIMATE$_{\mathscr{G}}(t, b, \boldsymbol{S})$ produces, and let $e_1, e_2, \ldots e_{t'}$ be the corresponding size estimates. Since TOPDOWNSIZEESTIMATE$_{\mathscr{G}}$ halts when the size estimate is at least $t$,

$$e_{t'} \geq t \quad \text{and} \quad e_{t'-1} < t.$$

We set $\Delta := \eta t$ and wish, for all $1 \leq j \leq t + \eta t$, that $e_j$ estimate the size of $T_j^\circ$ to accuracy $\pm\Delta$. Since the size of $T_j^\circ$ is $j$, we equivalently wish for

$$|e_j - j| \leq \Delta \quad \text{for all } j = 1, \ldots, t + \eta t. \tag{9}$$

Each $T_j^\circ$ has max depth at most $\log t + \log\log t$. By Lemma C.1 and a union bound over all $t + \eta t$ different $j$, we can guarantee that Equation (9) holds with probability at least $1 - \delta$ if we set

$$b \geq \frac{(2^{\log t + \log\log t})^2}{2\Delta^2} \cdot \ln\left(\frac{2t(1+\eta)}{\delta}\right)$$

$$= \Omega\left(\frac{(t \log t)^2}{(\eta t)^2} \cdot \log\left(\frac{t}{\delta}\right)\right)$$

$$= \Omega\left(\frac{(\log t)^2}{\eta^2} \cdot \log\left(\frac{t}{\delta}\right)\right).$$

Therefore, for the $b$ we set in Theorem 5, Equation (9) holds with probability at least $1 - \delta$. For the remainder of this proof, we suppose it holds and then show that the $t' \in [t - \eta t, t + \eta t]$. We first show that $t' \leq t + \eta t$. By Equation (9), for $j = t + \eta t$,

$$e_{(t+\eta t)} \geq (t + \eta t) - \Delta$$
$$\geq t.$$

Recall that $t'$ is the lowest integer such that $e_{t'} \geq t$. Therefore, $t' \leq t + \eta t$. We next show that $t' \geq t - \eta t$. By Equation (9) for $j = t' \leq t + \eta t$,

$$t' \geq e_{t'} - \Delta$$
$$\geq t - \eta t.$$

Therefore, Equation (9) implies $t' \in [t - \eta t, t + \eta t]$ proving that with probability at least $1 - \delta$.

$\qquad\qquad\qquad\qquad\qquad\qquad\qquad\qquad\qquad\qquad\qquad\qquad\qquad\qquad\qquad\qquad\qquad\qquad\square$

Finally, we show that the following algorithm estimates learnability.

---

EST$_{\mathscr{G}}(t, b, S^\circ, S_{\text{test}})$:

Return

$$\frac{1}{|S_{\text{test}}|} \sum_{(x,y) \in S_{\text{test}}} \mathbb{1}\left[\text{LOCALLEARNER}_{\mathscr{G}}(t, b, S^\circ, x) \neq y\right]$$

---

Figure 5: EST$_{\mathscr{G}}$ takes as input a size parameter $t$, a minibatch size $b$, and an unlabeled dataset $S^\circ$ and labeled test set $S_{\text{test}}$. It outputs the error of the tree returned by MINIBATCHTOPDOWN$(t', b, S)$ with respect to $S_{\text{test}}$, where $S$ is the labeled version of $S^\circ$ and $t'$ is close to $t$. As in Footnote 2, the random outcome of $\boldsymbol{B}_{\text{strands}}^\circ$ and the minibatches should be consistent across all runs of LOCALLEARNER$_{\mathscr{G}}$.

**Theorem 6** (Formal version of Theorem 3). *Let $f : \{\pm1\}^d \to \{0, 1\}$ be a target function, $\mathscr{G}$ be an impurity function, $S^\circ$ be an unlabeled training set, and $S_{\text{test}}$ be a labeled test set.*

*For all $t \in \mathbb{N}$ and $\eta, \delta \in (0, \frac{1}{2})$, if the minibatch size $b$ is as in Theorem 5, then with probability at least $1 - \delta$ over the randomness of $\boldsymbol{B}_{\text{strands}}^\circ$, EST$_{\mathscr{G}}(t, b, S^\circ, S_{\text{test}})$ labels*

$$q = O(|S_{\text{test}}| \cdot b \log t + b^2 \log t)$$

*points within $S^\circ$ and returns*

$$\text{error}_{S_{\text{test}}}(T) := \Pr_{(\boldsymbol{x}, \boldsymbol{y}) \sim S_{\text{test}}}[T(\boldsymbol{x}) \neq \boldsymbol{y}],$$

*where $T$ is as in Theorem 5.*

*Proof.* Based on Theorem 5, $\text{EST}_{\mathscr{G}}$ returns the desired result, so we only need to prove it labels few points within $S^{\circ}$. As in Footnote 2, the same $\boldsymbol{B}^{\circ}_{\text{strands}}$ are chosen across multiple runs of $\text{LOCALLEARNER}_{\mathscr{G}}$. As shown in Lemma D.1, the total number of points it labels is $O(b \log t) * m$, where $m$ is the number of strands built. $\text{EST}_{\mathscr{G}}$ needs to build $b$ strands for points within $\boldsymbol{B}^{\circ}_{\text{strands}}$ and $|S_{\text{test}}|$ strands for the points within $S_{\text{test}}$. As long as it caches its labels across runs of $\text{LOCALLEARNER}_{\mathscr{G}}$, the total labels used will be

$$q = O(b \log t) \cdot (b + |S_{\text{test}}|) = O(|S_{\text{test}}| \cdot b \log t + b^2 \log t).$$

$\square$