[Reviews · NeurIPS 2020]

Review 1

Summary and Contributions: This is a theoretical paper that shows that commonly used top-down decision tree learning heuristics are amenable to highly efficient learnability estimation. That is, for monotone target functions, it is possible to estimate the error the learned hypothesis will have on the training set S by observing just a small fraction of S.

Strengths: Nice, clean paper with very clear and interesting (but not surprising) results. Potentially interesting technique. Very well written

Weaknesses: The results are not surprising at all. That does not mean that it’s easy to prove, but it is still not surprising that it is possible to estimate how well a learning algorithm will do on a set S by observing only a small part of the set S. Also, the paper makes strong monotonicity assumptions, but does not discuss the implications of it on the strength (and relevance to application) of the results.

Correctness: Seems correct.

Clarity: Very well written.

Relation to Prior Work: Discussed well.

Reproducibility: Yes

Additional Feedback: The paper is not interesting enough for a competitive conference. It is good to have these results in the literature, but I suggest to send it to a journal. Having read the reviews, and following the discussion, I still think that this does not below in a competitive conference. Indeed, as the authors stress in their response, the power of the result is due to the specific algorithm developed here. Nevertheless, I cannot be excited by it, given the monotonicity assumption and the fact that it applies only to the uniform distribution setting. I agree that it's an interesting result, but I think that it's not interesting enough nor important enough for a top conference.


Review 2

Summary and Contributions: The paper presents applications of recent work on classical decision tree learning algorithms. Very efficient (sublinear) algorithms are given for estimating the error of the tree produced by a top-down learning algorithm (specified by an impurity function) for a monotone target function, using a given training set.

Strengths: The paper belongs to a flurry of beautiful recent work on the classical problem of learning decision trees using top-down algorithms. Results of Blanc et al. are strengthened and the strengthened result is applied to very natural new learning problem, estimating the quality of the hypothesis produced by a learning algorithm on a fixed training set. The proofs are interesting and elegant.

Weaknesses: There are a few comments for improving the presentation, given below.

Correctness: Yes.

Clarity: One point that would be useful to clarify is that most of the positive results on decision tree learning apply to the uniform distribution setting. This could be complemented by mentioning some general negative results for decision tree learning, and discussing whether there are possibilities to extend the results, for example, for product distributions. In the remark after Theorem 3 on arbitrary test distributions, the difference between the test distribution and the uniform distribution used to get T should be discussed.

Relation to Prior Work: Yes.

Reproducibility: Yes

Additional Feedback: The author response addressed the comments in a satisfactory manner.


Review 3

Summary and Contributions: The paper studies the recently introduced "estimated learnability" framework in the context of decision trees. More specifically, they analyse top-down decision tree learning heuristic and show that their expected preformance (in terms of error probability) can be computed with exponetially fewer examples than it would take to learn the DT. On the way they introduce a minibatch version of the standard top-down decision tree learning heuristic that comes with the same error guarantees but can be learned more efficiently by having to consider only polylog examples at every step.

Strengths: The paper makes a significant contribution to the recently proposed framework of estimating learnability. The obtained analysis also provides a deeper understanding of the workings of well-known and prominent decision tree learning heuristics. The obtained results are non-trivial and surprising.

Weaknesses: Since estimating learnability is a very novel field, it is at this stage impossible to say how important it will be in the future. However, the field seems to have already gained some traction and the main results were published at very strong venues.

Correctness: yes

Clarity: Given that the paper is very theoretical and the proofs (in the appendix) are very technical, I was happy to see that the authors made a very good effort to explain the main ideas behind their results in a very intuitive manner in the main paper. So yes the paper is well-written.

Relation to Prior Work: yes

Reproducibility: Yes

Additional Feedback:


Review 4

Summary and Contributions: The paper is a theoretical contribution in the domain of estimating learnability. It improves over results in [BLT20a] on learning monotone target functions with top-down decision tree algorithms. For this, they introduce a mini-batch top-down decision tree learning algorithm. They also provide additional results for active local learning and estimating learnability.

Strengths: A sound paper in the domain of computational learning theory for the recently introduced problem of learnability estimation.

Weaknesses: The focus is on sample complexity, more precisely on the number of examples to be labeled. My main concern is about the overall complexity. I would have liked to read more details on how the batch samples can be efficiently drawn at each leaf to be queried.

Correctness: The claims seem correct modulo the above question.

Clarity: The paper is well written.

Relation to Prior Work: The distinction with [BLT20a] could have been made more thorough.

Reproducibility: No

Additional Feedback: **** Thanks for the feedback. I do not change my evaluation on the paper. The main contribution is the mini-batch learning algorithm, but the batch sampling procedure needs to process the whole dataset. Therefore I am not convinced that the paper is enough for NeurIPS. I suggest to submit the paper to a conference or journal on computational learning theory.

[Author Response · NeurIPS 2020]

We thank the reviewers for their thoughtful and valuable feedback. We appreciate their time and effort, especially given the current uncertain times.

**Response to Reviewer 1:** We begin by responding to Reviewer #1's remark that the notion of estimating learnability is interesting but unsurprising:

"*The results are not surprising at all. That does not mean that it's easy to prove, but it is still not surprising that it is possible to estimate how well a learning algorithm will do on a set $S$ by observing only a small part of the set $S$.*"

There are many other algorithms for learning decision trees, based on generic algorithmic paradigms such as polynomial regression [LMN93, KKMS08] and bottom-up construction [EH89, MR02]. We in fact believe that for these other approaches, it is impossible to estimate learnability with the sample complexity achieved in this work: exponentially smaller than the information-theoretic minimum required for learning. This highlights a unique advantage of the top-down algorithms that we study in this work: one can build a tiny part of the hypothesis corresponding to a specific input, without constructing the entire hypothesis.

Regarding the notion of estimating learnability more generally, although it is still relatively new, there is already a growing body of work (appearing at recent NeurIPS, COLT, and AISTATS conferences; see lines 41-48 of our submission for references), studying it for a variety of learning problems. These works highlight novel connections between this notion and other areas of interest in both the theory (sublinear time algorithms, property testing, etc.) and practice (data selection, hyperparameter tuning, etc.) of machine learning. Our work is the first is to study this notion in the context of decision tree learning.

"*Also, the paper makes strong monotonicity assumptions, but does not discuss the implications of it on the strength (and relevance to application) of the results.*"

We thank the reviewer for raising this point. The focus of our work is on formal performance guarantees, and such guarantees for top-down algorithms are only known for monotone target functions. There are simple examples of non-monotone target functions for which top-down algorithms fare very poorly in the sense of building a tree that is no more accurate than a trivial classifier (unless we allow them to grow a huge tree). Monotonicity is a natural way of excluding these adversarial functions, and for this reason it is one of the most common assumptions in learning theory. Results for monotone functions tend to be good proxies for the performance of learning algorithms on real-world datasets, which also do not exhibit these adversarial structures. Just as ID3 and CART do, we expect our algorithm will work well in practice for most real-world datasets, even if they are not perfectly monotone. We will revise our paper to discuss this.

**Response to Reviewer 2:** We thank Reviewer #2 for suggestions for improving our presentation. We agree with them, and will incorporate these suggestions in our next revision.

**Response to Reviewer 4:** Regarding Reviewer's #4 point about the distinction between our work and [BLT20]: that work focuses on proving that top-down heuristics successfully learn monotone functions, whereas our focus is different. We have access to an unlabeled dataset, and wish to estimate how well those top-down heuristics would perform on the labeled dataset by only labeling a few points. Our design and analysis of mini-batch top-down is in service of our main goal, which is to give an algorithm for the aforedescribed learnability estimation task.

We thank the reviewer for their question about overall complexity. The runtime of our algorithm can be upper bounded by the product of the size of the dataset and the sample complexity of our learnability procedure. In particular, taking a batch sample from a particular leaf can be done in a single sweep through the dataset to determine which inputs are consistent with the leaf and then randomly sampling one of them. We will revise our paper to incorporate the runtime.

# References

[BLT20] Guy Blanc, Jane Lange, and Li-Yang Tan. Provable guarantees for decision tree induction: the agnostic setting. In *Proceedings of the 37th International Conference on Machine Learning (ICML)*, 2020.

[EH89] Andrzej Ehrenfeucht and David Haussler. Learning decision trees from random examples. *Information and Computation*, 82(3):231–246, 1989.

[KKMS08] Adam Kalai, Adam Klivans, Yishay Mansour, and Rocco A. Servedio. Agnostically learning halfspaces. *SIAM Journal on Computing*, 37(6):1777–1805, 2008.

[LMN93] Nathan Linial, Yishay Mansour, and Noam Nisan. Constant depth circuits, Fourier transform and learnability. *Journal of the ACM*, 40(3):607–620, 1993.

[MR02] Dinesh Mehta and Vijay Raghavan. Decision tree approximations of boolean functions. *Theoretical Computer Science*, 270(1-2):609–623, 2002.


[Meta-Review · NeurIPS 2020]

The submission got four reviews that were quite polarised in their recommendations, with two against accepting and two strongly in favour. The disagreement did not concern the technical quality of the paper. The reviewers agree that the theoretical work in this paper has been very competently performed and in the context of the problem the authors consider, the results are interesting and advance the state of the art. The disagreement is over whether the results are significant enough for NeurIPS or would be more appropriate for a specialised theory conference. The main objections against accepting are (i) the results are not surprising, (ii) the assumptions (monotonicity and uniform distribution) are strong and (iii) the overall computational complexity is high. Regarding (i), the authors in their rebuttal provide a reasonable discussion about why it is not obvious that such results should hold. Regarding (ii), the rebuttal points out and two reviewers agree that such strong assumptions are necessary for meaningful analysis of heuristic top-down algorithms (which otherwise can be shown to break down completely on some simple target classifiers); furthermore, such assumptions have traditionally been accepted as necessary for certain types of deep theoretical analysis. Regarding (iii), the rebuttal clarifies the result of the paper but does so in a direction that confirms the reviewer's negative evaluation. All the reviewers participated in discussion after the rebuttal, but opinions did not converge. Since the reviewers are strongly divided and the issue is largely a judgement call on the significance of the results, I'm making my recommendation based on my personal view, which is in favour of accepting the paper. On issues (i) and (ii), I tend to agree with the rebuttal by the authors and the comments of the two reviewers recommending accepting. On issue (iii) I'm a bit more hesitant, but I consider the results on label efficiency interesting enough. However, the rebuttal mentions that "the runtime of our algorithm can be upper bounded by the product of the size of the dataset and the sample complexity of our learnability procedure." I hope this is an off-the-cuff upper bound that can be improved be more careful use of data structures; otherwise I'm pessimistic about using the algorithm on (large) real-wold data sets which the authors mention.